# A Review of Estimation Methods for Aboveground Biomass in Grasslands Using UAV

Clara Oliva Gonçalves Bazzo [1],* , Bahareh Kamali [1], Christoph Hütt [2] , Georg Bareth [2] and Thomas Gaiser [1]

1 Institute of Crop Science and Resource Conservation (INRES), University of Bonn, Katzenburgweg 5, 53115 Bonn, Germany
2 GIS & RS Group, Institute of Geography, University of Cologne, Albertus-Magnus-Platz, 50923 Cologne, Germany
* Correspondence: clarabazzo@uni-bonn.de

**Abstract:** Grasslands are one of the world's largest ecosystems, accounting for 30% of total terrestrial biomass. Considering that aboveground biomass (AGB) is one of the most essential ecosystem services in grasslands, an accurate and faster method for estimating AGB is critical for managing, protecting, and promoting ecosystem sustainability. Unmanned aerial vehicles (UAVs) have emerged as a useful and practical tool for achieving this goal. Here, we review recent research studies that employ UAVs to estimate AGB in grassland ecosystems. We summarize different methods to establish a comprehensive workflow, from data collection in the field to data processing. For this purpose, 64 research articles were reviewed, focusing on several features including study site, grassland species composition, UAV platforms, flight parameters, sensors, field measurement, biomass indices, data processing, and analysis methods. The results demonstrate that there has been an increase in scientific research evaluating the use of UAVs in AGB estimation in grasslands during the period 2018–2022. Most of the studies were carried out in three countries (Germany, China, and USA), which indicates an urgent need for research in other locations where grassland ecosystems are abundant. We found RGB imaging was the most commonly used and is the most suitable for estimating AGB in grasslands at the moment, in terms of cost–benefit and data processing simplicity. In 50% of the studies, at least one vegetation index was used to estimate AGB; the Normalized Difference Vegetation Index (NDVI) was the most common. The most popular methods for data analysis were linear regression, partial least squares regression (PLSR), and random forest. Studies that used spectral and structural data showed that models incorporating both data types outperformed models utilizing only one. We also observed that research in this field has been limited both spatially and temporally. For example, only a small number of papers conducted studies over a number of years and in multiple places, suggesting that the protocols are not transferable to other locations and time points. Despite these limitations, and in the light of the rapid advances, we anticipate that UAV methods for AGB estimation in grasslands will continue improving and may become commercialized for farming applications in the near future.

**Keywords:** photogrammetry; grassland monitoring; precision agriculture; biomass estimation; vegetation indices; effective workflow



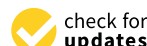

## 1. Introduction

Grasslands are among the largest ecosystems on the planet, playing an important ecological and economic role and contributing to the food security of millions of people [1]. According to FAO [2], grasslands cover 25% of the terrestrial surface, equivalent to around 68% of the world's agricultural areas. This makes grasslands an important provider of ecosystem services in different parts of the world [3,4]. When properly managed, grasslands can effectively contribute to carbon sequestration and improve air and water quality, nutrient cycling, and biodiversity, as well as food production [3,5,6].

Grasslands store 30% of the world's terrestrial biomass [7]. Moreover, the provision of aboveground biomass (AGB) is one of the most important ecosystem services in grasslands and constitutes the basis for increasing fodder productivity [8]. Thus, a precise and rapid method for the estimation of AGB is critical for the management and protection of grasslands [9–11] and for enhancing the sustainability of these ecosystems [12].

Current approaches to estimating AGB can be classified as either ground-based or remote sensing (RS) methods. Ground-based methods can be either destructive or non-destructive. Destructive methods traditionally involve cutting the grass in the field, followed by drying and weighing it in the laboratory [13]. Although these measurements generate the most accurate estimates of grassland biomass, they are time-consuming and labor-intensive [14].

Ground-based methods for non-destructive measurement of grassland AGB have been studied for decades [15,16]. These approaches estimate AGB using equations relating biomass to measurable biophysical factors such as plant height and plant density [17]. Handheld devices are the most straightforward instruments for measuring these biophysical factors [18]. The most widely used and well-documented ground-based method for the non-destructive measurement of AGB in grasslands is the rising plate meter (RPM) [19]. These instruments measure compressed sward height by integrating sward height and density over a specific area [20]. The ability of RPM-based compressed sward height to estimate AGB grass using regression models is now well established [21–23]. In view of this, farmers use RPM devices to create electromechanical models, which produce accurate and reliable estimates [24].

Despite the benefits of fast and regular assessments, the RPM method also has drawbacks, including operator variability and paddock slope. Through uneven and undulating terrain, the RPM method's ability to measure grass height effectively can be impacted, frequently leading to inaccurate measurements due to the RPM base not effectively touching the true ground surface [25]. The RPM also presents limitations when the sward is high and lacks a flat top structure, or when the grass sward is sparse and grows poorly and unevenly [26]. It is also not suitable for grasses with tender erect stems, including some tropical grasses [27]. Additionally, RPM measurements are also point measurements, and therefore, the within-paddock spatial variability of grassland biomass production is not taken into account because only an average paddock estimate is observed [28].

In recent years, RPM devices have become more sophisticated as technology has advanced. Ultrasonic distance sensors are used in devices such as the GrassHopper (TrueNorth Technologies, Shannon, Ireland) and the GrassOmeter (Monford AG Systems Ltd., Dublin, Ireland) [18]. In addition to handheld devices, vehicle-mounted devices have also been developed. Examples are the Pasture Meter (C-Dax Agricultural Solutions, Palmerston North, New Zealand) and the Pasture Reader (Naroaka Enterprises, Narracan, Australia). These sensors can monitor grass height while driving the vehicle through the center of a towing tunnel, where optical sensors detect grass height, which is then calibrated to estimate AGB [24].

Despite the benefits of fast and regular assessments provided by these sensing systems, there are still several drawbacks. In particular, the precise estimation of AGB in large-scale grassland ecosystems is difficult due to (1) limited spatial coverage, especially for handheld equipment, hence limiting the within-field description of the variability of the sward, (2) the requirement for heavy technical equipment, (3) limited access to the field due to grazing animals, (4) potential disturbances at a greater frequency for repeated measurements for vehicle-mounted sensors, and (5) applicability restrictions based on field conditions (e.g., soil moisture) [18,24].

RS-based methods offer potential for rapid and automated measurements to quantify both structural and biochemical properties of the vegetation at high spatial and temporal resolution at a range of spatial scales [18]. These methods include digital imaging (hyperspectral, multispectral, optical (red–green–blue, RGB), radar), photogrammetry, laser scanning, and combinations of various sensors on different platforms [29]. Numerous

studies have evaluated the feasibility of using satellite RS to estimate plant parameters. Although satellite platforms offer an effective way to collect data over large areas [14], using satellite imaging for calibrating and validating an AGB estimation model in grasslands may be inefficient due to low spatial resolution [8]. Most satellite systems with high spatial resolution (<5 m) are commercially operated, and therefore, image acquisition costs for short revisit times can become a limiting factor [30]. In a fragmented agricultural landscape, as seem in some grassland fields, where the average field size is low, high-spatial-resolution images are required [31]. Additionally, the applicability of satellite imagery can be significantly hampered and negatively impacted by weather conditions (cloud cover obstructing free sight) [32].

In recent years, unmanned aerial vehicles (UAVs), also known as remotely piloted aircraft systems, unmanned aircraft systems, or drones, have proven to be an important and viable tool for measuring and estimating biophysical parameters at a scale appropriate to grassland distribution [31]. With flexibility, UAVs can be operated quickly, simply, and economically. Most importantly, they can collect imagery data at high spatial, spectral, and temporal resolutions at exactly the point in time when the information is needed. In fact, when surveying objects at small (5 ha) to medium (5–50 ha) spatial scales, UAV-based photography outperforms alternative imaging acquisition technologies, such as satellites and manned aerial systems. Specifically, in this context, UAVs show higher temporal and spatial resolution as well as exhibit greater versatility at a lower cost [33].

In the past ten years, the number of research articles describing UAV applications has increased dramatically, with these studies encompassing a diversity of UAV types and applications [34]. More recently, there has been increased interest in applying UAV remote sensing to the estimation of AGB in grasslands. In this context, structural features of grasslands have been used for the estimation of grassland height and AGB [8,18,24]. Nevertheless, image-based approaches using UAV to estimate forage biomass are still in their infancy [35–37]. In view of this, there is no standard process for planning, collecting, and analyzing these data in order to extract AGB information. Considering the grassland's inherent properties, several aspects linked to data collection and analysis methodologies, as well as the study species and study site, can affect the accuracy and prediction of the resulting models. The methods often used to estimate AGB in grasslands by UAV imagery are similar to those used to monitor arable crops [38]. However, arable crops generally show lower heterogeneity than grasslands. Grasslands often exhibit substantial spatio-temporal heterogeneity due to highly diverse floristic compositions and co-occurrence of different phenological stages [18]. This heterogeneity affects the assessment of AGB in grasslands using UAVs [39]. AGB estimation in grasslands may be inaccurate or imprecise if these aspects are not taken into account.

A comprehensive review of the different methods and factors influencing the AGB estimation in grasslands is therefore essential to understand how each stage of the process affects outcomes so that subsequent data collection and analysis can produce accurate and reliable data. Although the utility of UAVs is well known in biomass estimation in agriculture, recently developed applications of UAVs to AGB estimation in grassland ecosystems have not yet been evaluated or systematically reviewed. To date, the majority of review studies of UAV for biomass estimation in agriculture have been broad, involving numerous fields and different remote sensing systems, and the description of biomass estimation with little emphasis on grassland-specific properties. To address this gap, we systematically review the use of UAVs in the estimation of AGB in grassland ecosystems. We perform a comprehensive literature review of the topic to (1) give an overview on common practices of the use of sensors, scale of work, ground truth methods, data processing, and analysis methods and (2) to identify which spectral and structural data are most accurate with respect to AGB estimation. We conclude by discussing the challenges and future prospects of UAV remote sensing in AGB estimation in grassland ecosystems.

## 2. Materials and Methods

Using the PRISMA protocol [40], we conducted a systematic review and meta-analysis of studies that use Unmanned Aerial Vehicles (UAVs) to estimate biomass in grassland systems. Figure 1 presents a flow diagram of the study selection process. In the identification step, relevant literature was retrieved from Google Scholar and Web of Science using search terms comprising keywords related to UAVs ("UAS", "UAV", "unmanned aerial system", "unmanned aerial vehicle") and to aboveground grassland biomass ("grass", "grassland", "pasture", "forage", "biomass", "aboveground biomass", "above ground biomass"). The search was limited to English-language research articles published from January 2011 to August 2022. We considered all types of grassland systems. This review did not consider studies classified as review papers, book chapters, reports, or Ph.D. theses.

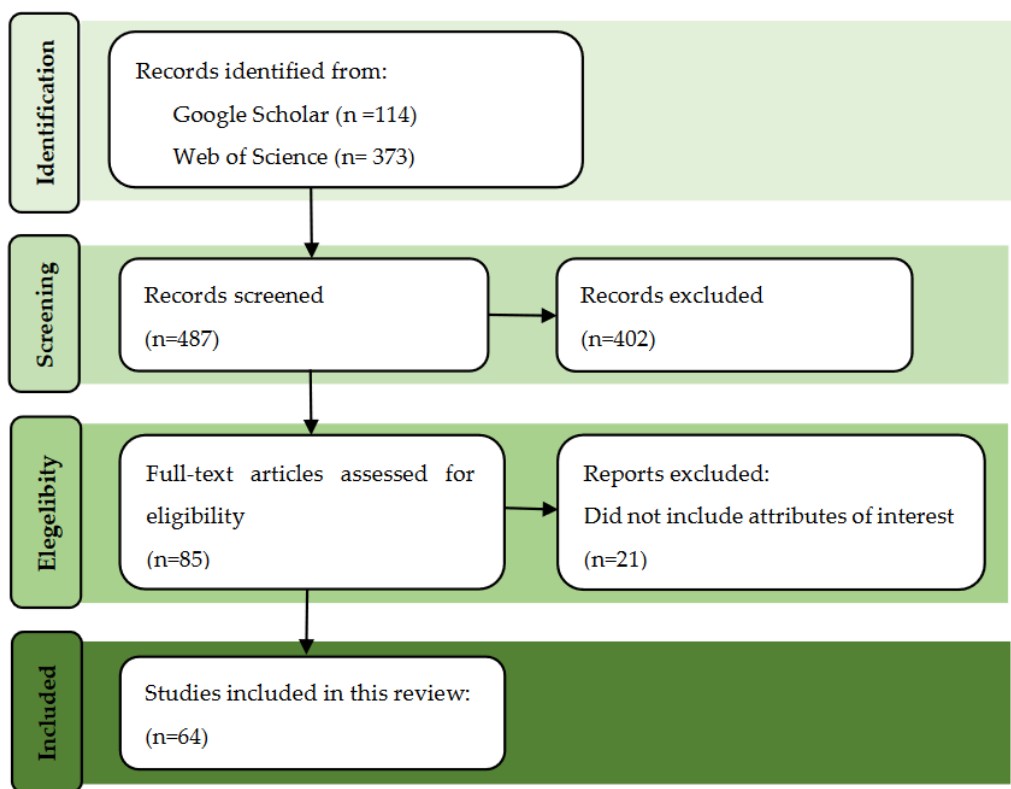

**Figure 1.** PRISMA flow diagram for study selection.

A total of 487 articles were obtained as a result of the Google Scholar and Web of Sciences searches. To be included in the review, a study was required to fulfill the following three criteria: (1) the study uses UAV and no other system type; (2) it focuses on grassland ecosystem; (3) it presents AGB estimation from UAV imagery. The articles identified in the first step were screened, and we consulted the title and abstract. After the screening phase, 85 research articles remained. We confirmed each study's eligibility by reading the full text, after which 21 studies were discarded because they did not contain extractable data for the following four features of interest: site attributes, biomass measures, UAV platform, and sensors. In total, 64 studies were retained, which had extractable data for all four features. For each article, we extracted metadata (Appendix A), including information related to the characteristics of the study site, grassland species composition, UAV platforms, flight parameters, sensors, field measurement, biomass indices, data processing, and analysis method.

## 3. Results and Discussion

An automated search of Google Scholar and Web of Sciences resulted in a final set of 64 papers that used UAV imagery to estimate AGB of grassland areas (Table A1). The

following sections provide a detailed description of meta-analysis findings, including general features of the articles and biomass estimation data analysis.

### 3.1. General Characteristics of Studies

Figure 2a presents the locations of the 64 studies considered in this review. In total, grasslands located in 15 countries were studied. Germany accounted for the largest number of studies (N = 14), followed by China (N = 10), the United States of America (USA) (N = 7), Australia (N = 5), Belgium (N = 4), Finland (N = 5), Brazil (N = 4), Estonia (N = 3), and Norway (N = 3). Studies in Canada, Ecuador, Ireland, Israel, Japan, Spain, South Korea, and Switzerland were represented by one publication each.

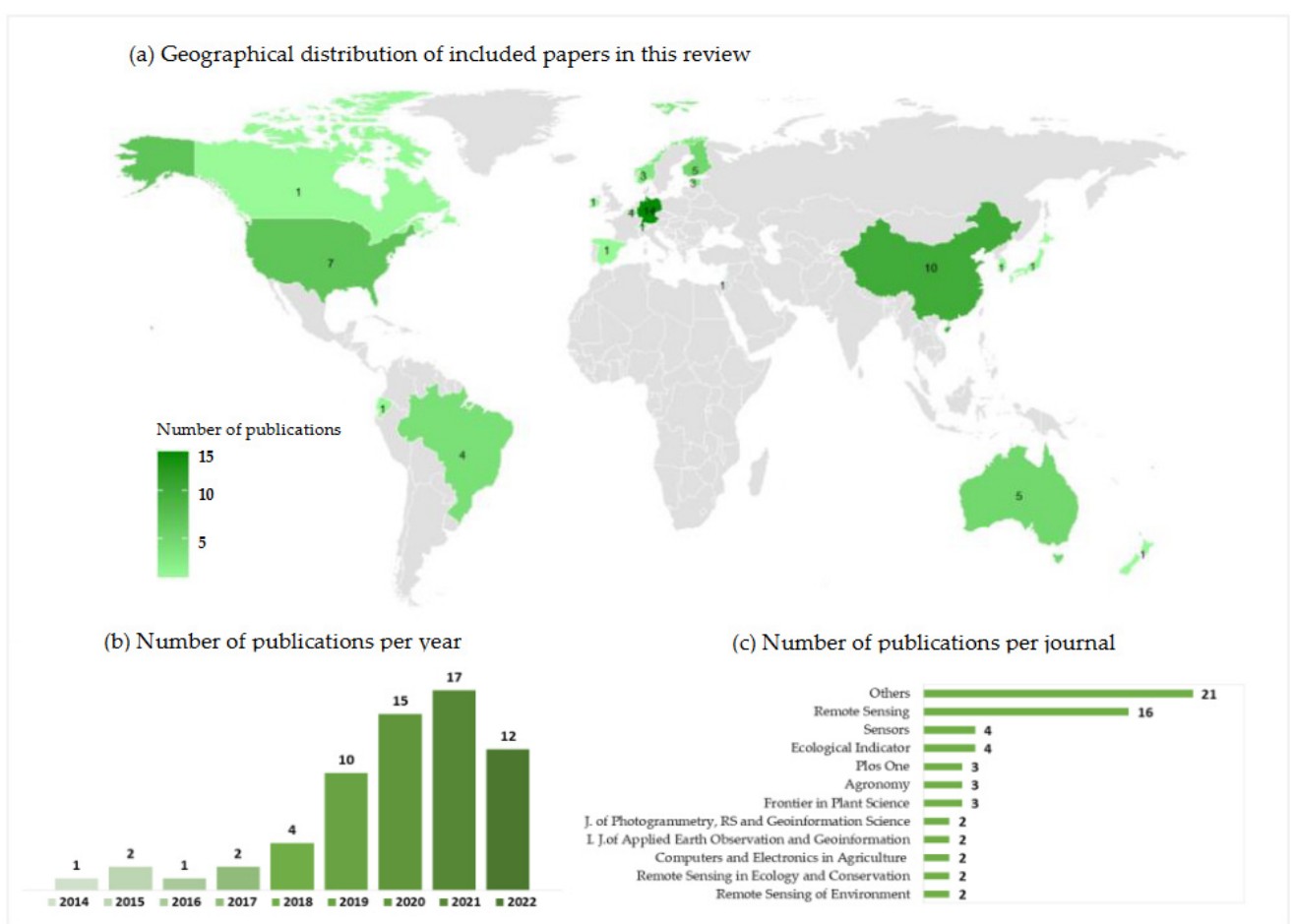

**Figure 2.** (**a**) Geographical distribution of included papers in this review. (**b**) Number of publications per year. (**c**) Number of publications per journal.

Figure 2b presents the number of articles published annually from 2012 to 2022. The first article, published in 2014 in the USA [41], used high-resolution imagery from a UAV to estimate biomass in a natural grassland site in the USA. From 2014 to 2017, only six papers were published, and subsequently, the number of publications increased steadily. Figure 2c shows only journals that published more than two papers. The most represented journals include *Remote Sensing* (16 papers), *Sensors* (4 papers), and *Ecology Indicator* (4 papers).

Considering the representation by continents, thirty-two of the sixty-four studies were conducted in European countries, twelve in Asia, eight in North America, seven in Oceania, and only five in South America, and no studies were conducted in the African continent. Although there are many significant areas of grassland in Europe and North America, which are often part of mixed farmland systems, much of the world's grassland area is located in the extensive natural grasslands of Central Asia, Sub-Saharan and

Southern Africa, North and South America, and Australia/New Zealand. Considering the scenario above, the productivity of journal articles about UAV applications for AGB biomass estimation in grassland regions with the largest representation of this vegetation worldwide is generally low. Studies should preferably be carried out in grassland biomes across several areas and continents [42]. More numerous and diverse grassland systems should be studied in order to improve UAV applications for AGB biomass estimation in grassland, particularly grasslands in regions that will be specifically impacted by climate change (e.g., tropical regions) [43], which are currently significantly under-represented in the available research survey.

### 3.2. Characteristics of the Study Sites

Regarding the characteristics of the study sites, 64 articles reported the type of grassland. Of these, 34 studies investigated fields as experimental sites, 18 investigated naturalized grasslands, and 12 investigated grassland farms. In addition, 62 publications reported whether the site included mono or multi-species grasslands. Of these, 46 publications studied multi-species grasslands, 15 studied mono-species systems, and 1 studied both systems (mono and mixed grasses). Fertilization conditions were described in 27 publications, of which only 3 studied organically fertilized grasslands. Animal presence in the grasslands was reported in 14 studies, of which 9 analyzed the effect of grazing activities on the biomass estimation.

The heterogeneity of the experimental site is an important feature since many studies suggest that increasing the species richness of grassland can reduce AGB estimation models' performance. According to Wijesingha et al. [44], biomass prediction for species-poor and homogenous grasslands had higher accuracy than biomass prediction for species-rich, diverse grasslands. Michez's et al. [45]'s results also suggest that the low species diversity in their experimental site (timothy-dominated pastures) probably improved the biomass modeling process. Grüner et al. [46] reported that the high variability of the canopy surface in legume–grass mixtures results in lower prediction accuracy compared with more homogeneous arable crops. They achieved $r^2$ values of 0.46 and up to 0.87 depending on the sward composition for mixed legume–grass swards and pure legumes and grass stands. Villoslada et al. [47] indicated similar trends in modeling accuracies, where sites characterized by the presence of more productive communities or a higher herbage yield show lower prediction accuracies than short-sward sites.

The distinct plant architectures in heterogeneous grasslands may have an impact on image acquisition due to poor modeling of plant extremities, resulting in a larger variability than monocultures and reflecting in lower $r^2$ values [48]. It has also previously been demonstrated that the complexity of sward structures, vegetation height, and plant species richness all influence the spectral properties of training samples [47]. The high heterogeneity in some grassland fields can also intensify the mixed pixel effect, an important remote sensing issue that affects the ability to monitor phenology [49]. This, in turn, influences the overall prediction accuracy. In addition, the potential for generalization of some studies is limited because they are based on approaches using site-specific data, which makes the relationships obtained difficult to transfer to other areas. Thus, study site selection should take into account local and regional variations, with the goal of incorporating a fair representation range of vegetation into the data collection process.

### 3.3. UAV Data Collection, UAV Data Processing, and Analysis Methods

In general, studies used a similar workflow to estimate AGB in grasslands using UAV data, as shown in Figure 3. Even though not all studies followed all of the steps, the standard process was adopted by many of the publications considered in this review. Typically, workflows included the following steps: (1) UAV imagery recording concurrent with ground control points (GCP) and ground-based field data collection; (2) UAV data processing, including pre-processing, creation of photogrammetric 3D point clouds and/or orthomosaics, georeferencing of point clouds and orthomosaics, creation of canopy height

models (CHM) using digital terrain models (DTM) and digital surface models (DSM) derivate from a digital elevation model (DEM), derivation of structural, textural, and/or multispectral, hyperspectral, or RGB spectral index; (3) generation of predictive AGB models using UAV-derived variables as predictors and ground-based AGB and/or CHM, and/or vegetation index. The overall goal of the next sections is to provide a comprehensive workflow description for AGB estimation in grasslands using UAV, with a specific focus on the main elements of the three steps: (1) field data collection, (2) image pre-processing, and (3) data analyses.

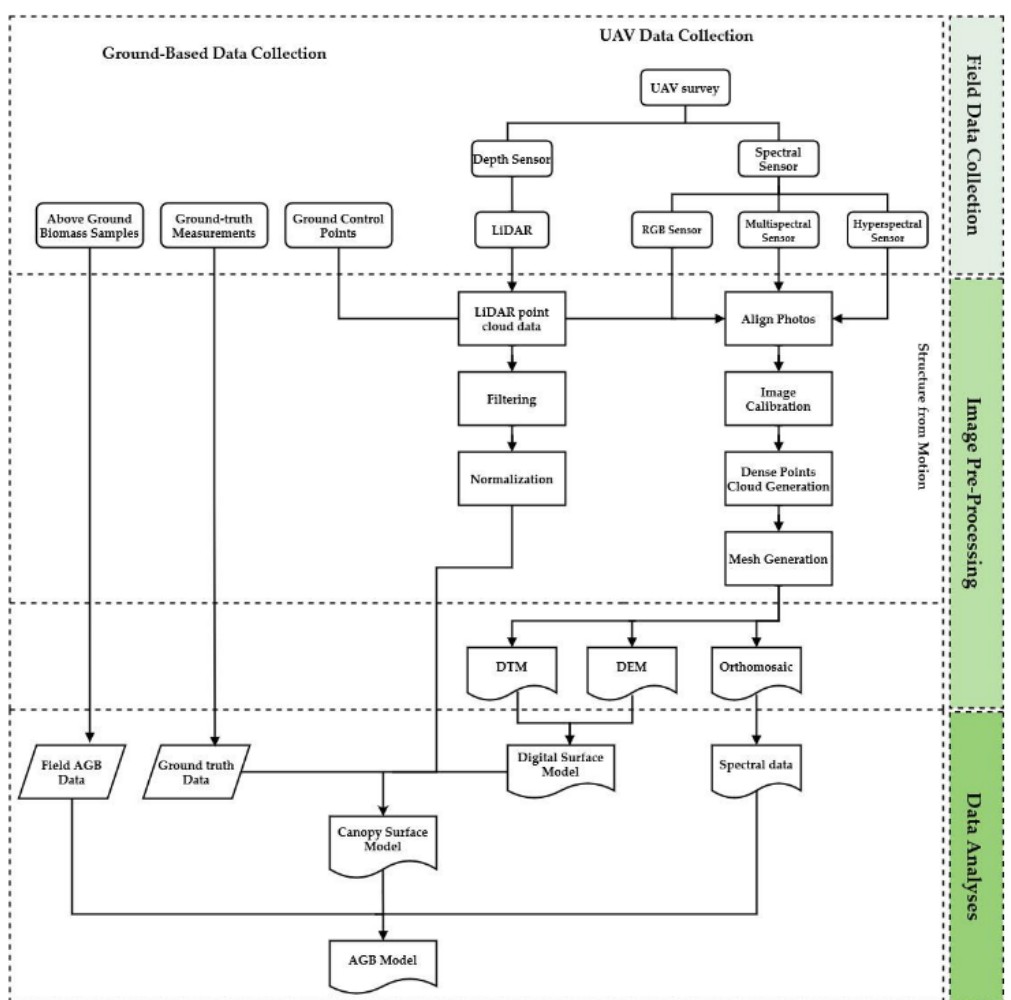

**Figure 3.** The data processing workflow by which grassland AGB model estimations are generated using Structure from Motion photogrammetry and LiDAR.

### 3.3.1. Field Data Collection

Data sampling as ground-based data collection and UAV flight is a critical step in AGB estimation. Some elements must be taken into consideration for an accurate data collection to ensure a reliable result. Table A2 presents a summary of field data information collected from the papers reviewed. Items recorded include location, type of field, type of grassland, number of sites, UAV platform, sensors, flight altitude, image front and side overlap, number of GCPs, ground sample distance (GSD), frequency of data collection, biomass ground truth, total number of biomass samples, biomass sample size, and canopy height measurement.

### Ground-Based Data Collection

Field measurements, such as biomass sampling and plant height measurements, are established methods for biomass estimation in grassland monitoring [50]. The quantitative

collection of data in the field is essential to establish, train, and evaluate biomass estimation models derived from UAV images. Additionally, in grassland ecosystems, the accuracy of canopy height and AGB estimation can be improved by using ground-measured data for calibration [51]. Grasslands typically have heterogeneous vegetation, and species contribution and yield vary in the field throughout the growing season, being influenced by different factors such as cutting intensity, soil management, and fertilization [48]. For reliable and precise biomass estimations in areas with such complex vegetation variety and high dynamics, sampling should be performed on a frequent temporal basis and with a large number of samples [52]. Morais et al. [53] reviewed the use of machine learning to estimate AGB in grasslands and concluded that the size of the field sampling is the most important factor to improve estimation accuracy, and increasing the size of the datasets should be one of the main priorities to improve the estimation models.

Regarding the frequency of the sampling, most parts of the studies performed only one field sampling (n = 25). The study of Borra-Serrano et al. [54] had the highest sampling frequency, with 22 collections in one year. The average number of field samples was 90, and the range was between 13 and 1403. According to Geipel et al. [55] the capacity of a model to perform well when applied to new scenarios improves with the size and variation of the calibration dataset, and many researchers have too small datasets to produce generalizable models. Qin et al. [56] concluded that, despite taking into account the spatial heterogeneity of AGB in vegetation patches, they are unable to validate the applicability of inversion results for each grassland type due to the small sample size. Capolupo et al. [57] also suggest that a larger and more representative training model sample size would improve model accuracy in their study. The intrinsic complexity and repeatability of field trial design, as well as the small sample size, were also constraints in the study of Lin et al. [58].

Compared to crops, the heterogeneous sward structure with high spatiotemporal variability in grasslands has the potential to alter the spatial distribution of biomass depending on the growth stage. As the results indicate, most studies use data from a minimal time span (e.g., a fraction of the growing season), limiting the ability to predict biomass in these complex and dynamic environments. When biomass prediction models are calibrated to the site, year, and even phenological stage of dominant plants, they become more robust [59]. In addition, the frequent collection of data over the course of the growing season could ensure that the dataset is diverse and that the models can be applied to various locations [42]. In this sense, Lussem et al. [16] recommended evaluating different swards under varying conditions and sites over multiple years. Pranga et al. [60] evaluated several growth periods, but the observation period was only one year with three cutting treatments. They also suggested that future research should incorporate data from other seasons/years, as well as different locations/conditions.

Regarding the AGB data collection method, samples were collected manually in 22 studies, mechanically in 20, and both methods in two studies. In seven studies, the method to collect samples was not specified, and in two studies, biomass samples were not directly collected but estimated by RPM calibration. There were two main procedures to sample AGB on the ground, collecting from quadrants or harvesting the entire plot. The sizes of the quadrants used for sampling varied between papers from 0.01 to 1 m$^2$. The most frequently used sizes were 0.25 m$^2$ (16 papers) and 1 m$^2$ (13 papers). The mechanical collection was the method used for all studies that sampling the entire plot and the size of the sampling ranged from 1 to 19.5 m$^2$.

Morais et al. [53]'s review concluded that the data collection procedure had a minimal impact on AGB estimation in grassland using machine learning methods. In their study, the average r$^2$ was lowest for the papers that used manual cutting (0.65) compared to mechanical harvesting (0.75). However, these findings are not statistically significant and are primarily a result of the different number of observations. We found comparable results, with an average r$^2$ for manual cutting of 0.68, which was lower than the r$^2$ observed for mechanical harvesting (0.82). These results can also indicate that the number of observations can have a greater impact on the accuracy of AGB estimation than the collection procedure.

In fact, similar to Morais et al. [53]'s results, we found that studies that employed manual cutting had both the lowest and greatest $r^2$ values (0.25 and 0.98). It should be noted, however, that the study with the lowest $r^2$ used 96 samples [61], whereas the study with the highest $r^2$ used 520 samples [62].

The plant cutting height is possibly a significant factor to take into account when collecting AGB samples in the field since it is challenging to cut vegetation right at ground level. Grassland biomass is distributed vertically in a pyramidal pattern, with increased biomass density closer to the ground [63]. In an Irish meadow, 40–60% of total biomass was distributed 0–10 cm aboveground, 30% was 20–30 cm aboveground, and less than 20% was more than 30 cm aboveground [64]. Only 17 of the studies included in this review reported the cutting height, which ranged from 2 to 10 cm above the ground. However, just two studies mentioned a height correction in the terrain model to compensate for the cutting height. In order to reduce the impacts of any residual stubble, Borra-Serrano et al. [54] used a correction factor of 5 cm to their baseline DTM. Karunaratne et al. [65] applied a constant offset of 7 cm to baseline DSM to compensate for the mowing height and pasture accumulation prior to the first measurement period. In this way, considering the distribution pattern of biomass in grasslands, we recommend that future models account for this factor to try to reduce discrepancies in reported results.

As for canopy height measurements, 29 studies did not mention the use of these data for biomass estimation. At least three studies mentioned the use of canopy height data in the field for biomass estimation but did not specify the data collection method. Of the 22 studies that used canopy height for biomass estimation, 11 used a ruler, tape, or height stick. In eight studies, the RPM was used to measure compressed canopy height. In three studies, field equipment such as a ground-based platform (PhenoRover) [66], Lidar Laser Scan [67], and the Rapid Pasture Meter (machine) [68] was used.

Most studies using SfM (Structure from Motion) to derive canopy height models for grassland have obtained reference measurements in the field with a height stick or a ruler and RPM since this equipment is more accessible and easier to use than mechanical equipment. However, because grassland plants differ significantly in canopy height, single or multiple tiller height measurements using manual methods would inevitably result in uncertainty about canopy height [69]. Bastitoti et al. [70] reported a high correlation between height measured with a ruler and a UAV with a multispectral sensor ($r^2$ = 0.89).The canopy heights estimated from UAV imagery and those measured using the ruler varied by about 8 cm. When comparing canopy surface models from UAV with manual reference measurements from height sticks, Grüner et al. [71] achieved $r^2$ values of 0.56 to 0.70 depending on the sward structure, species composition, and growing stage, while Viljanen et al. [26] report $r^2$ values of 0.61 to 0.93. Zhang et al. [51] also found that even though LiDAR-derived canopy height was lower than the ground-measured data, it showed a strong correlation with the height measured with a ruler ($r^2$ = 0.92). Wang et al. [72] reported that when compared to ground data measured with a ruler, LiDAR consistently overestimated the canopy height.

Because it effectively analyzes both canopy height and density, RPM is one of the most frequently used techniques for physical measurements of grassland sward height and the assessment of standing biomass [73]. Bareth et al. [50] report $r^2$ of 0.89 between RPM measurements and UAV-derived sward height. According to Lussem et al. [18], the performance of low-cost UAV-derived DSMs for estimating forage mass varies ($r^2$ = 0.57–0.73) depending on the harvest cut, but RPM measurements outperform the UAV model. However, canopy density, architecture, and plant developmental stage limit the accuracy of linear connections between RPM-based measurements and biomass. The results of some studies suggest that the agreement between the RPM and the UAV-borne equipment for measuring canopy height varied depending on canopy height and that the agreement was negatively impacted by low and high canopy heights in general [26,54,74]. RPM measurements demonstrated lower accuracy in sparse swards or tall, non-uniform canopies but better accuracy in dense swards and when the canopy has reached a height of 20–30 cm [26].

This inconsistency could be caused by the compression of the pasture induced by the RPM and canopy closure at high canopy heights. In the case of low canopy heights, this inconsistency may be caused by the ground being visible in the images, which reduces the digital surface model as a result of the photogrammetry software's point cloud interpolation. Considering this, RPM seems more suitable for measuring low grasses in their early phases of development.

Despite the significance of ground truth data for AGB model estimations, it is critical to remember that the available methodologies for measuring AGB and canopy height ground-based can also be subjective [75]. In addition, usually, ground truth data are either measured at a few locations in the field or at a single point on a plot and therefore do not necessarily provide a complete representation of the region of interest. In this way, in order to improve the validity of the ground-measured biomass data, it is important to take into account the limitations of the method and the biases of over- or under-estimate canopy height and AGB.

UAV Platforms

Multirotor platforms were the most commonly used UAV systems in the reviewed studies (87.66%), among which the quadcopter was the most widely deployed (58.46%) (Figure 4). In a review of studies on the use of UAVs and machine learning for agro-environment monitoring, Eskandari et al. [76] reported that fixed-wing models were the most used between 2015 and 2018. However, from 2018 to 2019, there was an increase in the use of quadcopter and hexacopter models, and these became the most used. Multirotor UAVs have increased in popularity since they are extremely versatile, with the ability to hover, rotate, and take images from nearly any angle. However, multirotor UAVs also present some disadvantages. Due to their vertical takeoff and landing and ability to hover, multirotor platforms demand more energy to fly, resulting in reduced sustainability and shorter flight periods [75]. If the survey height is low, backwash from the rotors may affect the vegetation being monitored by producing plant movements [77]. Multirotors are sometimes associated with inadequate Global Positioning System (GPS) receivers, which can lead to decreased position accuracy, particularly in hilly places where GPS coverage is limited [76]. When compared with fixed-wings, the most significant disadvantage of rotor UAVs is their short range and flight time [78]. Fixed-wing aircraft tend to have a faster top speed, a longer flying time, and a greater range than rotorcraft. Fixed-wing systems are useful for collecting data across broad areas for these reasons. Nonetheless, fixed-wing aircraft have less mobility and require more landing space.

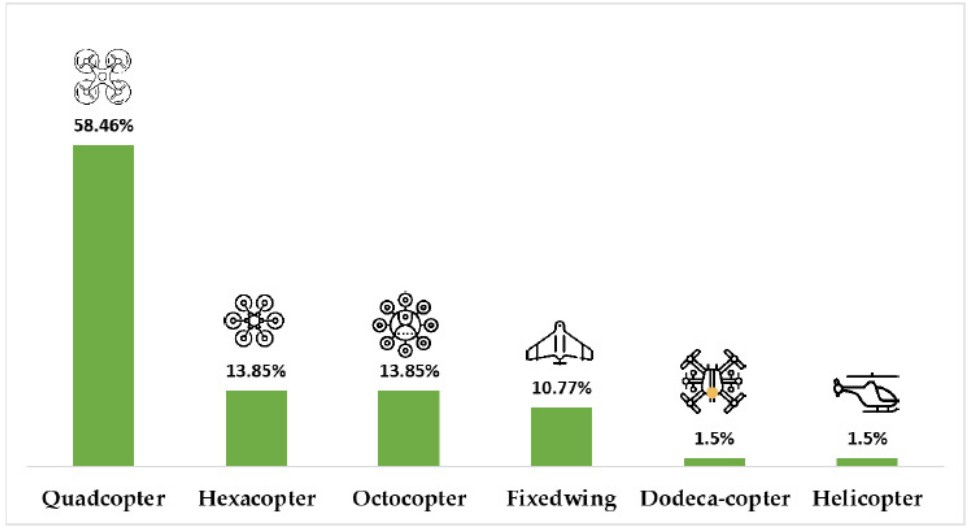

**Figure 4.** UAV platform types utilized per article.

According the review study of Poley and McDermind [75], there is no consistent difference in the accuracy of the biomass estimate model between studies using fixed-wing and multirotor platforms, and the selection of UAV platform depends on the research objective. A fixed-wing could be suitable if the study area is extensive, as in natural grasslands or larger grassland fields. A multirotor would be preferable for smaller and more challenging places, such as small grasslands and heterogeneous fields, where detailed vegetation imagery from a more stable platform is required.

Flight Parameters

Multiple interconnected elements during the UAV flight influence the quality of UAV-based outputs and, consequently, the AGB estimation. Because the precision of the output terrain data is determined by the accuracy of estimating tie points—and as a result, the reconstructed surface geometry—flight altitude is an important parameter. Reduced flight height results in smaller coverage areas, an increase in the number of flight missions required for a specific study site, and potentially increased variability in environmental conditions (e.g., cloudiness, sun angle), which complicates radiometric adjustment and decreases spectral accuracy. On the other hand, increasing altitude shortens flight time and allows one to cover larger areas, which can be important for maintaining relatively constant environmental conditions during the flight mission [79]. Higher altitude flights produce sparser point spacing, resulting in a less detailed DSM. For low-altitude flights, the result is a more irregularly shaped DSM, and these effects must be considered [80].

The 64 studies reviewed here deployed UAV flights at altitudes ranging from 2 to 120 m. The two flights with the lowest altitudes of 2 m were carried out in two studies by Zhang et al. [8,81] that evaluated the use of high-resolution images in generating quadratic models. The highest altitude flight (120 m) was carried out by Wang et al. [72] in a study testing if the relationship between tallgrass AGB measurements and spectral data is constant at different image spatial resolutions associated with different flight altitudes. The modal value for UAV altitude was 50 m (23% of studies), followed by 30 m (16%), 20 m (14%), 120 m (10%), 40 m (8%), less than 10 m (8%), 100 m (6%), 25 m (3%), 70 m (3%), 35 m (2%), 80 m (2%), 140 m (2%), 75 m (1%), 110 m (1%), 115 m (1%), and 120 m (1%).

Considering that plants and particularly grass leaves can be as thin as 2 cm, a higher spatial resolution may improve texture resolution and, as a result, biomass prediction accuracy. In the studies addressed in this review, most of the flights were performed at altitudes considered low (less than 100 m), with the most commonly used altitude being 50 m. Wang et al. [41] reported that surveying at 5 m above the canopy was more accurate than surveying at 20 or 50 m above the canopy in a tallgrass prairie ecosystem. Grüner et al. [82]'s study with different flight heights of 50 and 20 m resulted in an image resolution of 2–4 cm, which then had to be resampled to 4.5 cm. These authors recommend that different ground resolutions should be avoided in future studies to keep unified conditions for data analysis. Viljanen et al. [26] employed 30 and 50 m flight heights to estimate AGB in a mixed grassland field. The results for the 30 m flights produced lower reprojection errors (0.53–0.58) than the 50 m flights (0.783–1.25). The flights from a 30 m flying height also provided slightly better 3D RMSE (2.7–2.9 cm) than the 50 m flying height (2.8–5.0 cm). Näsi et al. [83] estimated grassland AGB using two flying heights of 50 and 140 m. Their study suggested that although employing datasets from 140 m produced promising results, adopting lower-height data can enhance AGB estimations.

The results obtained by DiMaggio et al. [48] indicate that flying at 50 m height can increase the area that is covered without considerably losing AGB estimation accuracy. The authors also recommended testing different altitudes to understand the relationship between pixel resolution and field data for AGB estimation. Karunaratne et al. [65] also evaluated the influence of different flight heights in their grassland AGB estimation models. The results indicate that the model generated at 25 m outperformed the other flying altitude models. However, the authors pointed out that, practically speaking, acquiring UAV data at a 100 m altitude provides a lot of benefits for farm-scale applications: (a) more coverage

of the land extent, (b) faster UAV data acquisition, and (c) smaller file sizes that allow for faster pre- and post-processing of collected datasets.

In this way, to establish best practice guidelines for using UAVs for on-farm applications and to adapt to changing technological advancements, it is also necessary to better understand the impacts of flying at various altitudes on the prediction quality of grassland AGB models. We also recommend that considering the specifications of the employed sensor, researchers should establish what GSD is necessary for identifying features of interest to AGB estimation. Then, in order to balance the necessary spatial resolution, tolerable error, and point cloud density with the most effective coverage of the study region, fly at the highest altitude where this GSD is possible [75].

The sequence in which the UAV flies also has an impact on data quality. The determination of forward and side image overlap is an important part of mission planning, especially for SfM photogrammetric reconstructions, which require features observed in multiple photos for building digital models, orthomosaics, and 3D models. The percentage of image overlap can affect the quality of the final SfM product, with more overlap leading to more precise final models. High overlap, on the other hand, necessitates the acquisition of more photos, increasing data volumes and computing time [79]. There are optimal overlap thresholds for specific vegetation types based on the surveyed area's specific characteristics and type of study. Agriculture fields and grasslands, which have low feature diversity and a relatively flat topography, demand a higher percentage of overlaps in order to extract tie points for the SfM algorithm [76]. Many studies examined in this review have employed considerable front and side overlap (median of about 80–70%). In the majority of the studies, a forward overlap of 80% and a side overlap of 60–75% resulted in high-quality orthoimages. The data are in agreement with the study by Eskandari et al. [75], which points out that the median for forward and side overlap is 80% and 70%, respectively, for UAV flights carried out in grasslands. Viljanen et al. [26]'s results also confirmed the main conception that the large image forward and side overlaps of approximately 80%, combined with self-calibration during photogrammetric processing, can provide a non-deformed photogrammetric block.

Sensor Technology

UAVs' ability to fly considerably closer to the ground than satellites or full-scale manned aircraft expands the range of sensors available and the spectral imaging. The spectral data obtained by an UAV can be simple RGB (red–green–blue) from an off-the-shelf camera or more specialized when employing multispectral, thermal, or even hyperspectral cameras. Among the studies reviewed here, visible sensors (RGB) are the most commonly employed sensor technology (48% of studies), followed by multispectral (29%), hyperspectral (16%), and LiDAR (Light Detection and Ranging) (7%) (Figure 5a). In terms of resolution, the sensors used in sensing can be classified as high resolution (between 0 and 10 cm), medium resolution (10 to 20 cm), and low resolution (more than 20 cm) [76]. The most commonly used data sources across the research are of high spatial resolution ranging from 0 to 10 cm (Figure 5b). Most studies (>80%) used data at high spatial resolution (0 to 10 cm), with visible and multispectral images being the preferred image types. Very few (<4%) studies used image data at low spatial resolution (>20 cm).

The increasing number of UAVs equipped with RGB commercial cameras has facilitated research using these low-cost sensors for grassland monitoring [16,24]. Compared to multispectral, hyperspectral, or thermal sensors, RGB sensors have a lower spectral resolution but a higher spatial resolution, and it is possible to calculate vegetation indices and estimate plant height from the same set of photographs. RGB sensors are also a more economical option, which is a significant benefit, especially for farm-scale applications.

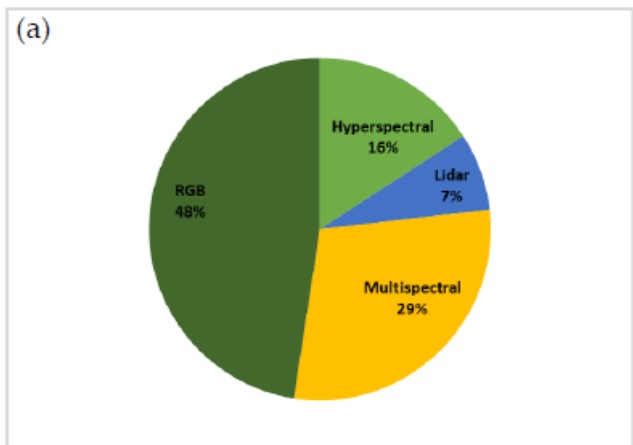
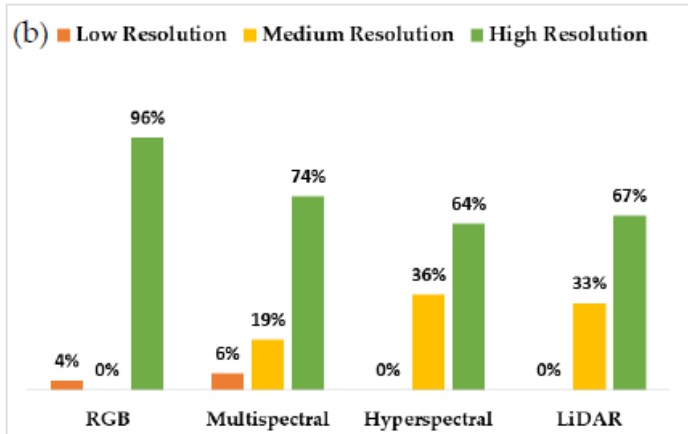

**Figure 5.** (**a**) Number of studies per sensor technology. (**b**) Image spatial resolution vs. sensor type.

Near-infrared (NIR) multi- and hyperspectral sensors have become more commonly accessible for UAVs over the past decade [30]. Initially, researchers used modified off-the-shelf RGB and near-infrared (NIR) cameras, as in the studies of Zhao et al. [9], Lee et al. [84], and Fan et al. [85]. These modified off-the-shelf RGB cameras were then replaced by specialized multispectral or hyperspectral cameras, which have decreased in cost and weight. Multispectral cameras, along with RGB cameras, are among the most commonly used sensors in the studies examined (30%). Multispectral sensors (e.g., MicaSense RedEdge 3 camera, Micasense, WA, USA) provide more spectral bands (e.g., red edge: 760 nm; near-infrared (NIR): 810 nm). The advantages of obtaining more spectral information for vegetation applications at an extremely high resolution collaborated for the increase in the use of multispectral sensors. The availability of a downwelling light sensor and radiometric calibration target are also key advantages of multispectral images. This allows the images to be radiometrically calibrated for repeatable and exact measurements less affected by environmental factors [60]. Hyperspectral sensors also measure reflectance in a wide range of spectral wavelengths. Such data are frequently processed into 3D point clouds utilizing Structure from Motion (SfM) procedures to offer information about the structure, texture, and variability of grassland areas [75]. This integration offers a lot of potential for accurate AGB estimation in grasslands. Especially when specific or many wavelengths are desired, multispectral and hyperspectral sensors can be used to obtain precise estimates of AGB. However, hyperspectral and multispectral sensors are still significantly more expensive than digital RGB cameras, which may be a drawback in farm-scale applications.

Even with the limitation on the spectral resolution range, the indices generated by RGB sensors can be cost-effective and have been applied in grassland for biomass estimation with acceptable or high levels of accuracy [16,18,83]. When evaluating several sensor types for detecting biophysical properties of vegetation, multiple studies discovered that RGB data from low-cost digital cameras produced AGB estimations comparable to or better than data from more expensive multispectral or hyperspectral sensors [18,46,71,83]. Few studies compared results from different sensors among the articles investigated for this review. However, in the studies that compared sensors, in most cases, there was no significant difference in accuracy in AGB estimation between RGB and other sensors. Lussem et al. [18] confirmed the potential of RGB techniques in AGB grassland modeling, achieving equivalent performance ($r^2 = 0.7$) using RGB or multispectral VIs. Näsi et al. [83] also stated that RGB can produce good results for AGB grassland modeling, even though it is inferior to the results of hyperspectral sensors. Compared with multispectral or hyperspectral imaging, the higher spatial resolution of RGB imagery could probably influence its ability to predict vegetation biomass more accurately [46,71].

The spectral resolution of UAV visible sensors is anticipated to continue to increase. Given the affordable prices, this platform will continue to be heavily utilized in AGB

estimates in grasslands. However, because the passive optical sensors mostly collect data from the top of the vegetation, there is little information available regarding the vertical structure of the vegetation, which limits the biomass estimate's accuracy. Another issue with optical imagery techniques is the possibility of natural light saturation when detecting high-density biomass plants.

Compared to optical sensors, LiDAR is an active remote sensing technology that can capture the vertical structure and height of vegetation as well as the three-dimensional coordinates of the target (Figure 6) [78]. LiDAR sensor is also unaffected by lighting conditions. In grassland ecosystems, UAV LiDAR has recently been employed to estimate canopy height and AGB. The study of Wang et al. [72] demonstrated that the LiDAR sensor has high potential for providing highly accurate grassland vegetation measurements, such as canopy height and fractional cover, which can then be used to estimate AGB on a large scale. The authors, however, pointed out that LiDAR alone would underestimate grassland canopy height and that field data calibration is required to achieve centimeter-level accuracy. Li et al. [69] concluded that incorporating LiDAR data considerably improved the performance of the spectral index in modeling and estimating AGB in grasslands in a non-destructive manner. Zhang et al. [51]'s results demonstrate that grassland AGB can be estimated using UAV LiDAR data under various grazing intensities.

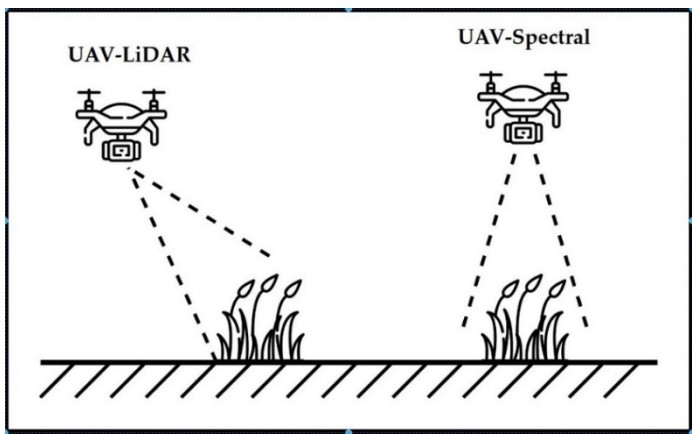

**Figure 6.** A schematic illustration of the difference between LiDAR and spectral data capture (adapted from Wang et al., 2021).

The study by Zhao et al. [61] indicates that, despite the tremendous potential for grassland AGB estimation, UAV LiDAR's sensor has a propensity to miss canopy data at canopy tops in grassland ecosystems. The canopy information loss can occur because UAV LiDAR collects data using a top-to-bottom view, and laser pulses may not completely penetrate the vegetation canopy. The challenge for UAV laser pulses to penetrate the canopy is further increased by the density of grassland vegetation, which may be, in some cases, much higher than in a forest [86]. According to Zhang et al. [51], the propensity of LiDAR sensors to not completely penetrate the high-density grassland canopy and the difficulty in receiving returns from the ground led to an underestimation of most canopy heights and the majority of fractional covers in the LiDAR data. As these attributes are used for AGB estimation, an underestimation in the data can lead to limitations in AGB estimation from LiDAR data.

Despite these limitations, LiDAR has been shown to outperform image-based techniques in terms of ground point capture and physical biomass parameter estimation [87], making it a promising technology for AGB estimation in grasslands. Nevertheless, in practice, the fact that commercial LiDAR sensors adapted for UAVs are still substantially more expensive than spectral sensors emphasizes the need to carefully evaluate the most cost-effective sensor for each specific aim.

3.3.2. Image Pre-Processing

The data collected by the UAV cannot be utilized directly to estimate biomass. In this way, a preprocessing step is usually included to guarantee that the data are suitable for further processing. Images taken from a UAV flight can be converted into 3D data using SfM-based software. Then, several objects can be classified from 3D data. Different companies have offered software solutions for processing photographs captured by UAVs, including functionalities for generating 3D spatial data for use in GIS (Geographic Information Systems) platforms, digital terrain and elevation models, generation of georeferenced orthophotos, and area and volume measurements. The different SfM software packages use different algorithms and processing options, which can affect the final outputs [88]. Of the 51 papers that mentioned the use of processing software, 52% (28 papers) used Agisoft Metashape (Agisoft LLC, St. Petersburg, Russia) to process UAV imagery data, followed by Pix4Dmapper (Pix4D, S.A., Lausanne, Switzerland) with 32% (17 papers). Furthermore, five papers employed other software, such as QGIS, ArcMap, and TerraScan.

None of the papers assessed in this review compared image preprocessing software, but previous studies have used Agisoft Metashape and Pix4Dmapper programs and evaluated the performance of both types of software. Kitagawa [89] captured characteristics from two experiments and compared them. Agisoft Metashape exhibited a clearer image but poor displacement extraction, whereas Pix4Dmapper had a z-value fluctuation but excellent displacement extraction. Isacsson [90] also examined the orthomosaic accuracies created from the same survey using Pix4Dmapper and Agisoft Metashape and also found that using Agisoft Metashape results in larger x and y position errors, whereas using Pix4Dmapper results in higher z error. Fraser et al. [91] compared the Agisoft Metashape and Pix4Dmapper software packages over a forested area of 235.2 ha. They concluded that Agisoft Metashape produced more detailed UAS-SfM outputs.

GCPs are high-visibility materials that are georeferenced using the Global Positioning System (GPS) after they are placed in a visible site to provide a point of reference for determining the position of the UAV in the area being photographed. By identifying GCPs with known coordinates visible in the imagery, a transformation that describes the relationship between the point cloud coordinate system and a real-world coordinate system can be used to georeference the point cloud that results from the reconstruction of SfM data [92]. Among the papers evaluated in this review, at least 62% (N = 33) mentioned the use of GCPs for the geometric correction of UAV images.

Reliable ground reference data are necessary for successful georeferencing [76]. Hence, the quantity and location of GCPs at the study site are crucial [75]. The geometric accuracy of surface and terrain models created from UAV imagery is likely to improve with more GCPs [44,54]. In the study of Wijesingha et al. [44], the small number of GCP was pointed out as a possible reason for the limitations of the DSM generated from the UAV data. The authors used only four GCP, which is the minimum for proper geo-referencing. They concluded that increasing the number of GCPs could increase the precision of SfM data and improve the model performance. It is also critical to place GCPs correctly [75]. The use of ground control points only around the edges of the study area rather than within plots can reduce the accuracy of surface and terrain models, so more GCPs should be placed throughout the entire area of interest [93]. Borra-Serrano et al. [54] reported that as grasses grew taller, GCP targets became more challenging to detect in imagery due to elongated plants. They recommended opening the area around the targets to guarantee they can be seen in all images throughout the growing season.

After the geometric correction step, the georeferenced sparse cloud is converted into a dense point cloud. The software computed the depth information by the image alignment for all points of the images. In the last step of pre-processing, the dense point cloud can be exported in the form of a DEM. DEMs are used to build a CHM of the grassland field (Figure 7). For this purpose, two types of DEMs are usually built: (1) DTM, corresponding to the ground, and (2) DSM derived from the imagery collected with the presence of canopy on the terrain.

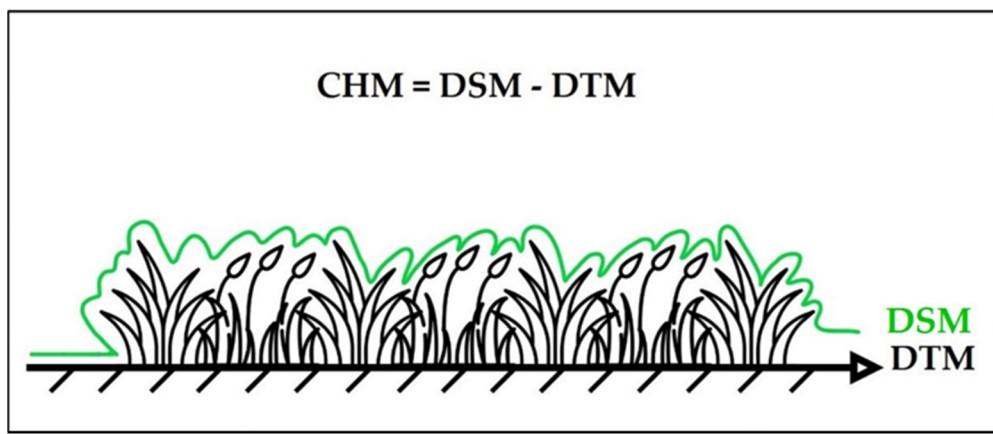

**Figure 7.** Graphical illustration of relation between digital surface model (DSM), digital terrain model (DTM), and canopy height model (CHM).

There are two main methods for extracting CHM information from UAV data. One method is to generate both DSM and DTM in raster format. This method considers the difference between DSM and DTM as the CHM [26]. This method is relatively simple, and since the analysis is carried out using raster analysis, the calculation is quick. However, applying interpolated DSMs and DTMs may lead to unwanted data smoothing. In the context of heterogeneous pasture growth, the use of such datasets could lead to the loss of information regarding the variability of CHM [65]. The second method uses the raw SfM point cloud dataset instead of an interpolated DSM raster. To determine CHM for every single point in the point cloud, the difference between the interpolated DTM raster of the study region is used [44]. Viljanen et al. [26] evaluated both methods and concluded that both provided similar DTMs and correlations to the AGB in grasslands.

A high-quality DTM with precise and accurate terrain representation is critical for extracting reliable estimates of vegetation structure from UAV imagery and therefore produce a reliable AGB estimation [75]. In areas with a dense vegetation canopy, such as some grassland fields, producing high-quality DTMs can be challenging [94]. Zhang et al. [8] pointed out that the density of grassland influenced the quality of DTM generated by an RGB sensor. An accurate DTM could not be produced because there were not enough ground points if the vegetation density was too high, but it was simple to extract ground points if it was moderate. In an ideal scenario, a reference DTM would be created beforehand when there is no vegetation, but this is not feasible, for example, in natural grasslands [95]. When a DTM is unavailable to represent the bare ground, point cloud classification is a frequently used technique to discriminate ground points (DTMs) and canopy height points (CHMs) from the same set of images [83]. Batistoti et al. [70] also found a solution by manually collecting GPS points to create the DTM, but this method is too time-consuming for large grassland fields. Alternatively, hybrid approaches combining SfM-derived DSMs with DTMs derived from LiDAR sensors have been explored [45]. Even so, special attention should be paid to potential errors in the LiDAR DTM, which is primarily based on ground point density and terrain variability. It is also important to notice that although producing input data such as DTM is a relatively simple task with LiDAR, the costs to obtain such products are high compared to RGB-only imagery [96].

### 3.3.3. Data Analyses

The ability of UAV image-derived models to accurately predict AGB is influenced by a variety of parameters connected to analysis methodologies. Table A3 summarizes the data analysis methods and essential results of the 64 papers considered in this review.

Most studies used statistical regression methods such as linear regression (LR), polynomial regression (PR), stepwise linear regression (SWL), multiple linear regression (MLR), and partial least squares regression (PLSR). LR was the most commonly used method

(n = 25). PLSR was used in 13 studies and MLR in 9 studies. Viljanen et al. [26] obtained the highest $r^2$ value ($r^2$ = 0.98) with MLR in a mixed grass experimental site using an RGB and HS sensor. The lowest $r^2$ value ($r^2$ = 0.25) was obtained by Zhao et al. [61] with SWL to estimate AGB in a mixed natural grassland field using a LiDAR sensor. Among the machine learning methods, random forest (RF) was the most popular and was employed by 16 studies. The highest median $r^2$ among all the papers assessed was obtained by Villoslada et al. [62] with RF and MLR (0.981), followed by Oliveira et al. [97] also using RF and MLR (0.97). RF has demonstrated competitive accuracy in biomass estimation when compared to other estimation methods used in agricultural applications [83]. Morais et al. [53] reviewed the use of machine learning to estimate biomass in grasslands. RF was also the method with the most applications, followed by PLSR.

The results of the different studies are highly variable and difficult to compare since they substantially depend on the type of grassland being monitored, the sensor (RGB, multispectral, hyperspectral, LiDAR), the usage of ground data, and 3D data. In the case of Zhao et al. [61] for instance, the lower result can be explained by the loss of canopy information in UAV LiDAR, which is an important factor influencing the estimation accuracy of AGB. Wang et al. [72] found comparable results ($r^2$ = 0.34) using a linear regression model and UAV LiDAR to estimate AGB in an experimental grassland site. On the other hand, Da Costa et al. [92] estimated AGB in a natural grassland using UAV LiDAR and simple linear regression but obtained a higher $r^2$ value ($r^2$ = 0.78).

Evaluating the result from different papers that use machine learning to estimate AGB in grasslands, Morais et al. [53] inferred that MLR has the greatest median $r^2$ (0.76), followed by PLSR (0.75) and RF (0.69). We found similar results evaluating the papers that informed the $r^2$ value for AGB estimation. Among the methods with more applications used in the papers evaluated in this review, RF has the greatest median $r^2$ (0.798). However, it differed slightly from the other methods, being followed by MLR (0.785), LR (0.78), and PLSR (0.776). Considering the small difference among the statistical methods, we agree with Morais et al. [53] that the accuracy of the analyses depends more on the quantity and quality of the data from field samples than on the type of statistical regression.

Among the papers assessed, at least 11 evaluated different regression methods for AGB prediction models using the same dataset. LR, MLR, and PLSR were commonly evaluated with other methods, probably because they are common regression techniques for predicting plant traits. Askari et al. [98] evaluated two regression techniques, PLSR and MLR, to estimate AGB using a multispectral sensor in a mixed grassland. The authors concluded that both PLSR and MLR techniques produced accurate models for AGB using only spectral data ($r^2$ = 0.77 and 0.76, respectively). The results from both techniques were considered robust enough to be employed, although the PLSR produced better model outputs. Comparing statistical methods for analyzing hyperspectral data from a grassland trial, Capolupo et al. [57] also found that PLSR was more effective at predicting AGB using specific vegetation indices. Lussem et al. [99] evaluated PLSR with other analysis methods, RF and support vector machine regression (SVR), with and without a combination of both structural (sward height; SH) and spectral (vegetation indices and single bands) features. In their study, however, the PLSR models were outperformed by the RF and SVR models. PLSR also was outperformed by other analysis methods (SVR, RF, and cubist regression (CBR)) in the study of Wengert et al. [100] using spectral data from a hyperspectral sensor.

Borra Serrano et al. [54] also evaluated PLSR with different regression models and one machine learning method (MLR, PCR, and RF) to estimate AGB using an RGB sensor in a monoculture grassland trial. Using spectral and structural data, MLR outperformed both the machine learning approach and other regression techniques in terms of AGB estimation. Geipel et al. [55] evaluated two regression methods, powered partial least squares regression (PPLSR) and LR, to estimate AGB using a hyperspectral sensor in a mixed grassland in an experimental site. Their results showed that PPLSR modeling approach fitted with reflectance data produced models with high AGB prediction accuracy

($r^2$ = 0.91). On the other hand, LR models using spectral indices and canopy height as predictor variables did not achieve satisfactory prediction accuracies.

Inputs

The selection of parameter(s) acquired from UAV data is probably the main element impacting the accuracy and prediction of AGB estimation in grasslands [75]. Spectral and structural (e.g., height) characteristics of grasslands are the most frequent inputs for predicting AGB using UAV data. Among the papers reviewed, 18 informed the use of only structural data as input, and 18 used only spectral data. Other 15 papers used both, while 11 papers used spectral and structural data combined with another data type. The study of Cunllife et al. [59] using canopy height and canopy volume as inputs had the highest $r^2$ (0.95) value among those that employed only structural data to estimate AGB. Among the studies that only used spectral data, Villoslada et al. [62] had the highest value for $r^2$ (0.98). For those studies that used both structural and spectral inputs, Oliveira et al. [97] obtained the best results ($r^2$ = 0.97) by evaluating different spectral indices and bands from a multispectral sensor, as well as eight canopy metrics from an RGB sensor. The mean $r^2$ value was 0.74 for studies that used only structural data, 0.77 for papers that only used spectral data, and 0.81 for papers that combined both structural and spectral data.

All studies that only employed structural measures used RGB and LiDAR data to generate metrics that represented the structure of the vegetation, and the most commonly used structural variable was canopy height. Some studies also used data such as vegetation volume, vegetation cover, and density volume factor. For vegetation with sparser or more varied canopies, such as grasslands, variables that reflect this heterogeneity, such as coefficient of variation, standard deviation, or percentiles of height, can be significant [75]. Zhang et al. [8] observed a significant correlation between AGB in a natural grassland and logarithmic regression using mean height derived from a UAV-RGB sensor ($r^2$ = 0.80). Wijesingha et al. [44] evaluated different canopy height metrics derived from a UAV-RGB sensor to estimate AGB in a mixed grassland farm. The results showed that among the canopy height metrics, the 75th percentile achieved the strongest explanatory power ($r^2$ = 0.63). Da Costa et al. [101] assessed different structural metrics derived from LiDAR data to estimate AGB from a natural grassland in the Brazilian savanna. The most accurate method employed metrics that represent canopy height (H98TH = height 98th percentile) and coverage (COV = cover percentage of first return above 1.30 m). For the estimation of AGB in a mixed natural grassland, Barnetson et al. [25] selected the maximum canopy height derived from a UAV-RGB sensor to closely approximate the settling height of the RPM measure.

The majority of studies that employed only spectral data used multispectral sensors (n = 9), followed by hyperspectral sensors (n = 3), RGB sensors (n = 3), and a fusion of different sensors (n = 3). These spectral datasets can be used as narrow bands or processed to derive a vegetation index (VI). VIs (ratios or linear combinations of bands) have been widely used in remote sensing research for vegetation identification, as they emphasize the differences in reflectance of the vegetation. The use of vegetation indices to characterize and quantify biophysical parameters of agricultural crops has two major advantages: (a) reducing the dimension of multispectral information through a simple number while minimizing the impact of lighting and target conditions, and (b) providing a number highly correlated to agronomic parameters. Several studies have found strong relationships between biomass measurements and RS-derived VIs [57,62,98,102]. Based on this relationship, a simple statistical methodology can be constructed to estimate plant biomass with the most suitable VI and optimal regression results.

Table A4 shows all 78 vegetation index formulations cited in at least one study for AGB estimation in grassland. Among the articles examined in this review, at least 38 used vegetation indices for biomass estimation analysis. Of the top five, the Normalized Difference Vegetation Index (NDVI) was the index used in most studies (N = 27), followed by Normalized Difference Red Edge (NDRE) (N = 16), the Green Normalized Difference

Vegetation Index (GNDVI) (N = 14), the Green Chlorophyll Index (GCI) (N = 10), and the Modified Chlorophyll Absorption in Reflectance Index (MCARI) (N = 9).

The results show a wide variety of indices, some of which might be more specific to certain indicators (e.g., Grassland Index, Plant Senescence Reflectance Index). However, most indices were used only once, and a few studies have compared the efficiency of multiple indices. The overall prevalence of NDVI was expected since this index is widely employed in various study scales to represent green vegetation abundance and net primary productivity in grasslands. Although it was the most used index and showed a good correlation for biomass estimation in a few studies [66,68], NDVI also has some limitations. NDVI presents sensitivity to the effects of soil brightness, soil color, atmosphere, and leaf canopy shadow and shows saturation in high-density vegetation. In fact, in some studies, NDVI did not perform better than preceding modeling strategies [60,95,103]. The study of Geipel et al. [104] showed that NDVI-based models appeared to be saturated at the first harvest dates and did not achieve an acceptable prediction level. This conclusion is similar to that of Karunaratne et al. [65] and Togeiro de Alckmin et al. [105], who suggested that predicting dry biomass only based on NDVI (as in previous studies) is ineffective. This is probably related to the saturation effect that occurs when the plant achieves higher levels of leaf area index. Indeed, Ref. [60] reported that with leaf area index (LAI) values larger than 3, NDVI exhibited a lower biomass estimation capability. EVI and GNDVI, on the other hand, saturate less at increasing LAI values and have been identified as significant predictive variables. In at least two studies comparing different vegetation indices to estimate biomass in grasslands, GNDVI performed better than NDVI [95,103].

Furthermore, it is important to consider a diverse set of vegetation indices in order to avoid the issues that come with less sensitive indices such as NDVI. When assessing various vegetation indexes, it is also critical to consider saturation, sensitivity, plant growth phases, canopy structure, and environmental impact [102].

Recently, several methods investigated the integration of different data by combining spectral and non-spectral data, and they found an improvement in the assessment of AGB in grasslands. According to the study of Lussem et al. [99], the combination of structural and spectral features can improve the estimation accuracy for AGB in grasslands. Viljanen et al. [26] reported that using MLRand RF to combine structural and spectral information resulted in a small improvement in AGB estimation. For the AGB estimation of perennial ryegrass in the study by Pranga et al. [60], the combination of spectral and structural characteristics from a multispectral camera utilizing random forest produced the best results. When combining vegetative indices and 3D features at various flight altitudes, Karunaratne et al. [65] observed a consistent improvement of AGB estimation.

The structural features, such as canopy height, were more significant for the AGB prediction models than the spectral features when both were combined [99]. Michez et al. [67] obtained an RMSE of 0.09 kg m$^2$ by combining VIs and canopy height and concluded that the canopy height had the highest significance in the multilinear regression model. Grüner et al. [82] developed AGB estimation by comparing RF and PLS models of spectral features with and without texture. They concluded that adding texture features improved the estimation models significantly. When predicting AGB using a fused dataset (from the RGB camera and the MS camera), Pranga et al. [60] likewise discovered that the canopy height characteristics were of the utmost significance; nevertheless, estimating the AGB with only the CH features produced rRMSE of 30–35%. Comparatively, the rRMSE of the AGB estimation was generally 10% lower.

It is important to note, however, that although these methods show promising results, combining spectral and non-spectral data in an applied setting can be more challenging because it requires employing several sensors or constructing complex data processing chains [95].

## 4. Challenges and Future Prospects

UAV remote sensing for AGB estimation in grasslands is still challenging, mainly due to the intrinsic characteristics of this ecosystem. The vegetation communities in grasslands are mainly composed of a variety of site-specific plant species that can contrast in size and phenology stage. Additionally, because grasslands are perennial, monitoring systems must be able to adapt to a wider variety of measuring conditions [52]. Future research should consider the inherent characteristics of these ecosystems, seasons, management practices, data collection parameters, and automation techniques in order to establish robust methods that can be transferred into management tools for grassland professionals [48]. We also strongly recommend that future studies provide more information on the agronomic aspect of the research area. A detailed overview of soil characteristics, spatial heterogeneity of species distribution, climate, grassland classification, and management practices used enables independent analyses and cross-study comparisons.

A significant constraint of UAV studies for AGB model estimation in grasslands is the low number of sampling intervals or limited representativeness due to the small number of sites and management intensities that can be assessed [100]. Furthermore, additional points must be explored. For example, because most studies only consider one growing season, future research could include more observations throughout different growing seasons. In this way, researchers will produce more high-quality datasets describing the temporal dynamics of vegetation in grassland ecosystems, which is recommended for improving AGB estimation models. Models created using a dataset based on numerous years, different management practices, and preferably multiple sites are more generalizable. As a result, they may better represent conditions at other sites and over different years [106]. Additionally, models should also be validated on a range of grassland fields from diverse locations and years to improve their practical applicability [100].

Apart from data collection, data processing and analysis are major factors in using UAVs for AGB estimation in grasslands. The processing of UAV data differs significantly from the processing of satellite data, creating a new demand for data processing software and suitable workflows. Additionally, image processing takes more time as spatial and spectral resolutions rise; therefore, more effective methods must be designed. Future directions for AGB grassland estimate may be accomplished by the ongoing reduction and cost-effectiveness of sensors, platforms, and computer hardware, as well as strong algorithms.

## 5. Conclusions

The present manuscript provides a comprehensive review of the most recent results in the field of UAV for AGB estimation in grasslands. Several factors can have a significant impact on the performance and generalizability of vegetation AGB estimation in grasslands throughout the data collection to data processing and analysis. Our findings are summarized as follows:

- The frequency of publications on grassland AGB estimation with UAV has increased over time and continues to rise, indicating the scientific community's interest.
- The frequency of studies is poorly distributed around the world, with South American and African grasslands appearing to be underrepresented. As a result, additional research should be conducted on some important grassland areas.
- The type of grassland, the heterogeneity, and the growth stage can strongly influence the AGB estimation model.
- Collecting ground-based data is a crucial step in estimating AGB in grasslands. The biomass sampling method seems to have a small influence on the accuracy of the AGB model estimation, whereas the number of samples is one of the main factors to improve the estimation accuracy.
- The measurement of canopy height is an important variable, especially for models that use structural data as input. However, the methods for collecting canopy height at the field level present limitations. RPM measurements demonstrated lower accuracy in sparse swards or tall, non-uniform canopies, and a measuring tape is based on an

"average height", but determined visually and rather subjective. The biases of each method must be taken into account to reduce inconsistencies in the results.

- Quadcopters were the most widely used platform, accounting for almost 60% of all platforms. Nevertheless, the type of platform has a low impact in AGB grassland estimation, and the selection of the platform depends more on the research objective.
- The modal value for UAV flight altitude among the studies was 50 m. Adopting lower altitude flights seems to enhance AGB estimations as this increase the spatial resolution. For farm-scale applications, however, collecting UAV data at higher altitude offers more advantages. We suggest flying at the highest altitude where the desirable GSD is possible.
- Large image forward and side overlaps of approximately 80%, combined with self-calibration during photogrammetric processing, can provide better data quality.
- In terms of sensor type, RGB was the most commonly employed (48%). Despite MS and HS sensor has the advantage to provide more spectral bands RGB data seems capable to produce models with comparable accuracy. In terms of cost–benefit and data processing simplicity, RGB sensors appear to be the most suitable for estimating AGB in grassland at the moment. The emergence of reliable and cost-effective LiDAR and hyperspectral sensors will have a significant impact on future research.
- For the reliable estimation of vegetation structure in grasslands from UAV imagery, a high-quality DTM with a precise and accurate representation of the terrain is necessary. However, UAV-derived DTMs may underestimate or overestimate field terrain differences depending on the canopy's density and the spatial resolution of the image.
- The accuracy of georeferencing models increases when a larger number of ground control points are equally distributed throughout the study area.
- Linear regression was the most commonly used regression model (n = 25). Random forest was the most popular machine learning method (n = 16). The findings suggest that the accuracy of the analysis methods is more dependent on the quantity and quality of data from field samples rather than the method itself.
- The most common inputs for AGB prediction in grasslands using UAV are spectral and structural data. Canopy height metrics were the most used structural data. At least 68% of the articles used vegetation indices for biomass estimation, with NDVI being the most commonly used. The results indicate that models that employed both data types (structural and spectral) outperformed models that only used one.

**Author Contributions:** Conceptualization, C.O.G.B., B.K., C.H., G.B. and T.G.; methodology, C.O.G.B.; formal analysis, C.O.G.B.; investigation, C.O.G.B.; writing—original draft preparation, C.O.G.B.; writing—review and editing, C.O.G.B., B.K., C.H., G.B. and T.G.; supervision, B.K., C.H., G.B. and T.G.; funding acquisition, T.G. and G.B. All authors have read and agreed to the published version of the manuscript.

**Funding:** This research was funded by the German Federal Ministry of Education and Research (BMBF) through the Digital Agriculture Knowledge and Information System (DAKIS) Project [Grant number 031B0729E] and the consortium research project "GreenGrass" [grant number 031B0734F], and partially funded by the Deutsche Forschungsgemeinschaft (DFG, German Research Foundation) under Germany's Excellence Strategy—EXC 2070—390732324 (PhenoRob).

**Data Availability Statement:** The data presented in this study are available in the Appendix A.

**Acknowledgments:** We would like to acknowledge writing consultation provided by Brian Cusack during Science Craft's Online Writing Studio.

**Conflicts of Interest:** The authors declare no conflict of interest.

## Appendix A

**Table A1.** Studies using UAV data to estimate grassland above ground biomass (AGB).

| No. | Title | Ref | Year | Journal | Main Objective |
|---|---|---|---|---|---|
| 1 | Modeling above-ground biomass in tallgrass prairie using ultra-high spatial resolution sUAS imagery | [41] | 2014 | *Photogrammetric Engineering & Remote Sensing* | To examine relationship between herbaceous AGB for the tallgrass prairie and its biophysical parameters derived from ultra-high-spatial-resolution imagery. |
| 2 | Estimating plant traits of grasslands from UAV-acquired hyperspectral images: a comparison of statistical approaches. | [57] | 2015 | *International Journal of Geo-Information* | To investigate the utility of hyperspectral images acquired from UAV for predicting vegetation traits in grasslands considering the plant phenology and fertilization on spectral data. |
| 3 | Mapping Herbage Biomass on a Hill Pasture using a Digital Camera with an Unmanned Aerial Vehicle System | [84] | 2015 | *Journal of The Korean Society of Grassland and Forage Science* | To develop a simple and cost-effective low-altitude aerial platform system with a commercial digital camera on an UAV system to collect images and estimate the herbage biomass using statistical analyses. |
| 4 | Ultra-fine grain landscape-scale quantification of dryland vegetation structure with drone-acquired structure-from-motion photogrammetry | [59] | 2016 | *Remote Sensing of Environment* | To develop a new technique to quantify biomass and associated carbon stocks in heterogeneous and dynamic short sward semi-arid rangelands. |
| 5 | Hyperspectral aerial imaging for grassland yield estimation | [104] | 2017 | *Advances in Animal Biosciences* | To investigate the potential of UAV imaging spectroscopy for in-season grassland yield estimation. |
| 6 | Modeling Aboveground Biomass in Hulunber Grassland Ecosystem by Using Unmanned Aerial Vehicle Discrete Lidar | [72] | 2017 | *Sensors* | To investigate if the canopy height, fraction cover, and aboveground biomass can be derived using models established from UAV-based discrete LIDAR data with desirable accuracy at quadrat and subplot scales. |
| 7 | Low-cost visible and near-infrared camera on an unmanned aerial vehicle for assessing the herbage biomass and leaf area index in an Italian ryegrass field | [85] | 2018 | *Grassland Sciences* | To demonstrate the use of a UAV system equipped with a low-cost visible and near-infrared (V-NIR) camera to assess the spatial variability in herbage biomass and LAI in an Italian ryegrass field. |
| 8 | Estimating biomass and nitrogen amount of barley and grass using UAV and aircraft based spectral and photogrammetric | [83] | 2018 | *Remote Sensing* | To develop and assess a methodology for crop biomass and nitrogen estimation, integrating spectral and 3D features that can be extracted using airborne miniaturized multispectral, hyperspectral, and color (RGB) cameras. |
| 9 | A novel machine learning method for estimating biomass of grass swards using a photogrammetric canopy height model, images and vegetation indices captured by a drone. | [26] | 2018 | *Agriculture* | To develop and assess a novel machine learning technique for the estimation of canopy height and biomass of grass swards utilizing multispectral photogrammetric camera data. |

**Table A1.** *Cont.*

| No. | Title | Ref | Year | Journal | Main Objective |
|---|---|---|---|---|---|
| 10 | Estimation of Grassland Canopy Height and Aboveground Biomass at the Quadrat Scale Using Unmanned Aerial Vehicle | [8] | 2018 | *Remote Sensing* | To develop a novel method for estimating the quadrat-scale aboveground biomass of low-statute vegetation. |
| 11 | Evaluation of grass quality under different soil management scenarios using remote sensing techniques. | [98] | 2019 | *Remote Sensing* | To evaluate the efficiency of hyperspectral and multispectral (UAV and satellite) remote sensing techniques for predicting and mapping grass biomass and crude protein under conventional grassland management in a temperate maritime climate. |
| 12 | Estimating pasture biomass and canopy height in Brazilian savanna using UAV photogrammetry. | [70] | 2019 | *Remote Sensing* | To estimate the canopy height using UAV photogrammetry and to propose an equation for the estimation of biomass of Brazilian savanna (Cerrado) pastures based on UAV canopy height. |
| 13 | Canopy height measurements and non-destructive biomass estimation of *Lolium perenne* swards using UAV imagery. | [54] | 2019 | *Grass and Forage Science* | To develop a methodology for monitoring the spatial and temporal dynamics of biomass accumulation of perennial ryegrass plots throughout the growing season in an affordable, easy-to-use, reliable, and non-destructive way using an RGB camera mounted on a UAV. |
| 14 | Biomass Prediction of Heterogeneous Temperate Grasslands Using an SfM Approach Based on UAV Imaging | [71] | 2019 | *Agronomy* | To develop of prediction models for dry matter yield in temperate grassland based on canopy height data generated by UAV RGB imaging over a whole growing season including four cuts. |
| 15 | Estimation of spatial and temporal variability of pasture growth and digestibility in grazing rotations coupling unmanned aerial vehicle (UAV) with crop simulation models | [68] | 2019 | *PLOS One* | To monitor, assess and manage changes in pasture growth, morphology, and digestibility by integrating information from an UAV and two process-based models. |
| 16 | Estimating biomass in temperate grassland with high resolution canopy surface models from UAV-based RGB images and vegetation indices | [18] | 2019 | *Journal of Applied Remote Sensing* | To evaluate the potential of low-cost UAV-based canopy surface models to monitor sward height as an indicator of grassland biomass and compare with established methods for biomass monitoring. |
| 17 | Mapping and monitoring of biomass and grazing in pasture with an unmanned aerial system | [67] | 2019 | *Remote Sensing* | To evaluate the potential of UAV as a tool for the characterization of pasture 3D structure (sward height) and aboveground biomass at a very fine spatial scale. |
| 18 | Comparing UAV-Based Technologies and RGB-D Reconstruction Methods for Plant Height and Biomass Monitoring on Grass Ley | [37] | 2019 | *Sensors* | To evaluate aerial and on-ground methods to characterize grass ley fields, composed of different species mixtures and estimate plant height, biomass and volume, using digital grass models, and avoiding the unnecessary destruction of the swards. |

Table A1. *Cont.*

| No. | Title | Ref | Year | Journal | Main Objective |
|---|---|---|---|---|---|
| 19 | Evaluating soil-borne causes of biomass variability in grassland by remote and proximal sensing | [73] | 2019 | *Sensors* | To investigate the relationship between soil characteristics and biomass production to identify high- and low-yielding regions within the field and their possible soil-borne causes. |
| 20 | Evaluation of 3D point cloud-based models for the prediction of grassland biomass | [44] | 2019 | *International Journal of Applied Earth Observation and Geoinformation* | To evaluate 3D point clouds derived from a terrestrial laser scanner (TLS) and an UAV-borne SfM approach for grassland biomass estimation over three grasslands with different composition and management practice in northern Hesse, Germany. |
| 21 | Estimating Plant Pasture Biomass and Quality from UAV Imaging across Queensland's Rangelands | [25] | 2020 | *AgriEngineering* | To demonstrate the use of UAV hyperspectral remote sensing to detect both crude protein and acid detergent fiber in a range of native pastures across the rangelands of Queensland, Australia. |
| 22 | Deep learning applied to phenotyping of biomass in forages with UAV-based RGB imagery | [96] | 2020 | *Sensors* | To propose a deep learning approach to estimate biomass in forage breeding programs and pasture fields using only UAV-RGB imagery and AlexNet and ResNet deep learning architectures. |
| 23 | A Pilot Study to Estimate Forage Mass from Unmanned Aerial Vehicles in a Semi-Arid Rangeland | [48] | 2020 | *Remote Sensing* | To develop a method to estimate forage mass in rangelands using high-resolution imagery derived from the UAV using a South Texas pasture as a pilot site. |
| 24 | Development and validation of a phenotyping computational workflow to predict the biomass yield of a large perennial ryegrass breeding field trial | [66] | 2020 | *Frontiers in Plant Science* | To validate a computational phenotyping workflow for image acquisition, processing, and analysis of spaced-planted perennial ryegrass to estimate the biomass yield of 48,000 individual plants through NDVI and plant height data extraction. |
| 25 | The potential of UAV-borne spectral and textural information for predicting aboveground biomass and N fixation in legume-grass mixtures | [82] | 2020 | *PLOS One* | To develop harvestable biomass and aboveground nitrogen fixation estimation models from UAV multispectral imaging of legume–grass mixtures with varying legume proportions (0–100%). |
| 26 | Comparison of Spectral Reflectance-Based Smart Farming Tools and a Conventional Approach to Determine Herbage Mass and Grass Quality on Farm | [107] | 2020 | *Remote Sensing* | To evaluate two spectral reflectance-based smart farming tools for determining herbage mass and quality of multi-species grasslands—a portable NIRS and a model to analyze multispectral imagery. |
| 27 | Investigating the potential of a newly developed UAV-based VNIR/SWIR imaging system for forage mass monitoring | [102] | 2020 | *Journal of Photogrammetry, Remote Sensing and Geoinformation Science* | To investigate the potential of a multi-camera system with a novel approach to extend spectral sensitivity from visible-to-near-infrared (VNIR) to short-wave infrared (SWIR) (400–1700 nm) for estimating forage mass from an aerial carrier platform. |

**Table A1.** *Cont.*

| No. | Title | Ref | Year | Journal | Main Objective |
|---|---|---|---|---|---|
| 28 | The fusion of spectral and structural datasets derived from an airborne multispectral sensor for estimation of pasture dry matter yield at paddock scale with time | [65] | 2020 | *Remote Sensing* | To develop empirical pasture dry matter (DM) yield prediction models using an UAV-borne sensor at four flying altitudes. |
| 29 | High-throughput switchgrass phenotyping and biomass modeling by UAV | [69] | 2020 | *Frontiers in Plant Science* | To exploit UAV-based imagery (LiDAR and multispectral approaches) to measure plant height, perimeter, and biomass yield in field-grown switchgrass in order to make predictions of bioenergy traits. |
| 30 | Monitoring Forage Mass with Low-Cost UAV Data: Case Study at the Rengen Grassland Experiment | [16] | 2020 | *Journal of Photogrammetry, Remote Sensing and Geoinformation Science* | To investigate the potential of sward height metrics derived from low-cost UAV image data to predict forage yield. |
| 31 | Can Low-Cost Unmanned Aerial Systems Describe the Forage Quality Heterogeneity? Insight from a Timothy Pasture Case Study in Southern Belgium | [45] | 2020 | *Remote Sensing* | To investigate the potential of off-the-shelf UAS systems in modeling essential parameters of pasture productivity in a precision livestock context: sward height, biomass, and forage quality. |
| 32 | Machine learning estimators for the quantity and quality of grass swards used for silage production using drone-based imaging spectrometry and photogrammetry | [97] | 2020 | *Remote Sensing of Environment* | To develop and assess a machine learning technique for the estimation of the quantity and quality of grass swards based on drone spectral imaging and photogrammetry. |
| 33 | An efficient method for estimating dormant season grass biomass in tallgrass prairie from ultra-high spatial resolution aerial imaging produced with small unmanned aircraft systems. | [42] | 2020 | *International Journal of Wildland Fire* | To investigate the viability UAV image data to estimate dormant season grassland biomass, based on the assumption that grassland canopy height correlates with grassland biomass. |
| 34 | Fine scale plant community assessment in coastal meadows using UAV based multispectral data | [47] | 2020 | *Ecological Indicators* | To assess the potential of UAVs and multispectral cameras for classifying and fine-scale mapping of plant communities in coastal meadows. |
| 35 | Using multispectral data from an unmanned aerial system to estimate pasture depletion during grazing | [28] | 2021 | *Animal Feed Science and Technology* | To develop and validate empirical models to estimate pasture depletion in paddocks while cattle are grazing using an UAV-borne multispectral sensor with rising plate meter measurements as the reference data. |
| 36 | Monitoring ecological characteristics of a tallgrass prairie using an unmanned aerial vehicle | [108] | 2021 | *Restoration Ecology* | To evaluate the potential applications of UAV derived data within restored tallgrass prairies using an affordable sensor and UAV. |

**Table A1.** *Cont.*

| No. | Title | Ref | Year | Journal | Main Objective |
|---|---|---|---|---|---|
| 37 | Predicting pasture biomass using a statistical model and machine learning algorithm implemented with remotely sensed imagery | [109] | 2021 | *Computers and Electronics in Agriculture* | To test the performance of an integrated method combining remote sensing imagery acquired with a multispectral camera mounted on an UAV, statistical models, and machine learning algorithms implemented with publicly available data to predict future pasture biomass loads. |
| 38 | Forage yield and quality estimation by means of UAV and hyperspectral imaging | [55] | 2021 | *Precision Agriculture* | To investigate the potential of in-season airborne hyperspectral imaging for the calibration of robust forage yield and quality estimation models. |
| 39 | Prediction of Biomass and N Fixation of Legume–Grass Mixtures Using Sensor Fusion | [46] | 2021 | *Frontiers in Plant Science* | To develop a multi-temporal estimation model for aboveground biomass and nitrogen fixation of two legume–grass mixtures. |
| 40 | The Application of an Unmanned Aerial System and Machine Learning Techniques for Red Clover-Grass Mixture Yield Estimation Under Variety Performance Trials | [103] | 2021 | *Remote Sensing* | To present a rapid, non-destructive, low-cost framework for field-based red-clover DM yield modeling. |
| 41 | A novel UAV-based approach for biomass prediction and grassland structure assessment in coastal meadows | [62] | 2021 | *Ecological Indicators* | To compare two temporal pre-harvest dry matter prediction capabilities under one- and two-year clover–grass cultivation fields with three different treatments and compare the performance of three machine learning algorithms and their corresponding variable importance rankings in estimating clover–grass mixture dry matter. |
| 42 | UAV Multispectral Imaging Potential to Monitor and Predict Agronomic Characteristics of Different Forage Associations | [110] | 2021 | *Agronomy* | To show a first screening of the potential of airborne multispectral images captured with UAVs for the monitoring and prediction of several in situ agronomic parameters of different forage associations by exploring the relationships between a few spectral indices UAV-based and simultaneous field measurements over several fields of forage associations. |
| 43 | Improving Accuracy of Herbage Yield Predictions in Perennial Ryegrass with UAV-Based Structural and Spectral Data Fusion and Machine Learning | [60] | 2021 | *Remote Sensing* | To examine the potential of UAV-based structural and spectral features and their combination in herbage yield predictions across diploid and tetraploid varieties and breeding populations of perennial ryegrass. |
| 44 | Effects of plateau pikas' foraging and burrowing activities on vegetation biomass and soil organic carbon of alpine grasslands | [56] | 2021 | *Plant and Soil* | To quantitatively assess the foraging and burrowing effects of plateau pikas on vegetation biomass and soil organic carbon at plot scale. |

Table A1. *Cont.*

| No. | Title | Ref | Year | Journal | Main Objective |
|---|---|---|---|---|---|
| 45 | Estimating dry biomass and plant nitrogen concentration in pre-Alpine grasslands with low-cost UAS-borne multispectral data–a comparison of sensors, algorithms, and predictor sets. | [94] | 2021 | *Biogeosciences Discussions* | To investigate the potential of low-cost UAS-based multispectral sensors for estimating aboveground biomass (dry matter) and plant community nitrogen concentration of managed pre-alpine grasslands. |
| 46 | Remote sensing data fusion as a tool for biomass prediction in extensive grasslands invaded by *L. polyphyllus* | [111] | 2021 | *Remote Sensing in Ecology and Conservation* | To develop prediction models from sensor data fusion for fresh and dry matter yield in extensively managed grasslands with variable degrees of invasion by *Lupinus polyphyllus*. |
| 47 | Improved Estimation of Aboveground Biomass of Disturbed Grassland through Including Bare Ground and Grazing Intensity | [112] | 2021 | *Remote Sensing* | To estimate alpine meadow AGB from multi-temporal drone images at a micro-scale and improve estimation accuracy in relation to two types of external disturbances (mowing-simulated grazing and rodents). |
| 48 | Biomass estimation of pasture plots with multitemporal UAV-based photogrammetric surveys | [106] | 2021 | *International Journal of Applied Earth Observation and Geoinformation* | To investigate the use of multitemporal UAV-based imagery and SfM photogrammetry to estimate the AGB of pastures at a fine spatial scale. |
| 49 | Remotely piloted aircraft systems remote sensing can effectively retrieve ecosystem traits of alpine grasslands on the Tibetan Plateau at a landscape scale | [113] | 2021 | *Remote Sensing in Ecology and Conservation* | To propose a framework for monitoring ecosystem traits by UAV visible remote sensing, verify the feasibility in monitoring ecosystem traits, quantify the contribution of each band in prediction, validate the prediction model, and generate high-spatial-resolution maps of ecosystem traits. |
| 50 | Estimation of forage biomass and vegetation cover in grasslands using UAV imagery | [95] | 2021 | *PLOS One* | To test and compare three approaches based on multispectral imagery acquired by UAV to estimate forage biomass or vegetation cover in grasslands. |
| 51 | Using UAV LiDAR to Extract Vegetation Parameters of Inner Mongolian Grassland | [51] | 2021 | *Remote Sensing* | To investigate the ability of Riegl VUX-1 to model the AGB at a 0.1 m pixel resolution in the Hulun Buir grazing platform under different grazing intensities. |
| 52 | Hyperspectral retrieval of leaf physiological traits and their links to ecosystem productivity in grassland monocultures. | [114] | 2021 | *Ecological Indicators* | To evaluate the remotely sensed retrieval of plant physiological traits and test the links between the intra- and inter-species trait variations and ecosystem productivity based on a grassland monoculture experiment. |
| 53 | A non-destructive method for rapid acquisition of grassland aboveground biomass for satellite ground verification using UAV RGB images | [81] | 2022 | *Global Ecology and Conservation* | To develop and assess the vertical and horizontal indices from UAV RGB images as predictors of grassland AGB at quadrat scale using the RF machine learning technique and verify whether the indices and methods are suitable for different grassland ecosystems over a large region. |

**Table A1.** *Cont.*

| No. | Title | Ref | Year | Journal | Main Objective |
|---|---|---|---|---|---|
| 54 | Analysis of UAV LIDAR information loss and its influence on the estimation accuracy of structural and functional traits in a meadow steppe | [61] | 2022 | *Ecological Indicators* | To investigate how UAV LIDAR information loss may occur and how it may influence the estimation accuracy of grassland structural and functional traits by comparing it with terrestrial laser scanning (TLS) and field measurements in a meadow steppe of northern China. |
| 55 | Estimation of aboveground biomass production using an unmanned aerial vehicle (UAV) and VENµS satellite imagery in Mediterranean and semiarid rangelands | [49] | 2022 | *Remote Sensing Applications: Society and Environment* | To develop a synergistic UAV and satellite imagery method to estimate AGB by integrating high-resolution UAV data with moderate resolution satellite data, and to assess AGB under different grazing pressures. |
| 56 | Beyond trees: Mapping total aboveground biomass density in the Brazilian savanna using high-density UAV-LiDAR data | [101] | 2022 | *Forest Ecology and Management* | To assess the ability of high-density UAV-LiDAR to estimate and map AGB across the structurally complex vegetation formations of the Cerrado in Brazil. |
| 57 | Quantification of Grassland Biomass and Nitrogen Content through UAV Hyperspectral Imagery—Active Sample Selection for Model Transfer | [52] | 2022 | *Drones* | To evaluate the use of UAV hyperspectral imagery for the quantification of forage yield and nitrogen nutrition status and implement and validate a supervised approach for model transfer. |
| 58 | Estimating Grass Sward Quality and Quantity Parameters Using Drone Remote Sensing with Deep Neural Network | [115] | 2022 | *Remote Sensing* | To investigate the potential of novel neural network architectures for measuring the quality and quantity parameters of silage grass swards, using drone RGB and hyperspectral images, and compare the results with the random forest (RF) method and handcrafted features. |
| 59 | Herbage Mass, N Concentration, and N Uptake of Temperate Grasslands Can Adequately Be Estimated from UAV-Based Image Data Using Machine Learning | [99] | 2022 | *Remote Sensing* | To estimate aboveground dry matter yield (DMY), nitrogen concentration (N%), and uptake (Nup) of temperate grasslands from UAV-based image data using machine learning (ML) algorithms. |
| 60 | Silage Grass Sward Nitrogen Concentration and Dry Matter Yield Estimation Using Deep Regression and RGB Images Captured by UAV | [116] | 2022 | *Agronomy* | To assess the suitability of CNN-based approaches by comparing different deep regression network architectures and optimizers to estimate grass sward nitrogen concentration (N) and dry matter yield (DMY) using RGB images collected from a drone. |
| 61 | Nitrogen variability assessment of pasture fields under an integrated crop-livestock system using UAV, PlanetScope, and Sentinel-2 data | [117] | 2022 | *Computers and Electronics in Agriculture* | To evaluate the spatial distribution of N in pasture fields cultivated under an integrated crop–livestock system (ICLS) using unmanned aerial vehicle (UAV) and satellite data. |

**Table A1.** *Cont.*

| No. | Title | Ref | Year | Journal | Main Objective |
|---|---|---|---|---|---|
| 62 | Effects of disturbances on aboveground biomass of alpine meadow in the Yellow River Source Zone, Western China | [118] | 2022 | *Ecology and Evolution* | To quantify the singular and combined effects of artificial grazing and pika disturbance severities on AGB and its changes in an alpine grassland on the Qinghai–Tibet Plateau, assessing the relative importance of both disturbances. |
| 63 | UAV-based prediction of ryegrass dry matter yield | [119] | 2022 | *International Journal of Remote Sensing* | To determine the accuracy of UAV-based prediction of percentage cover, vegetation volume, and DM yield in autumn from ryegrass sub-plots and compared to the current manual practice of harvesting, drying, and weighing. |
| 64 | Multisite and Multitemporal Grassland Yield Estimation Using UAV-Borne Hyperspectral Data | [100] | 2022 | *Remote Sensing* | To develop and evaluate UAV-based models with the goal of forage yield estimation of eight grassland habitats along a gradient of management intensities. |

**Table A2.** A summary of data field collection from papers assessed in the review.

| Reference | Local | Type of Field | Type of Grassland | Number of Sites | UAV Platform | Sensors | Flight Altitude (m) | Overlap, Side Overlap (%) | GCP | GSD (cm/Pixel) | Frequency of Data Collection | Biomass Ground Truth Data | Total Number of Biomass Samples | Biomass Sample Size (m²) | Canopy Height Measurement |
|---|---|---|---|---|---|---|---|---|---|---|---|---|---|---|---|
| (Alvarez-Hess et al., 2021) [28] | Australia | Grassland Farm | Mono | 2 | Quadcopter | MS | 50 | 80/80 | 10 | n/a | 2 collections in one year | RPM calibration | 529 | n/a | RPM |
| (Adar et al., 2022) [49] | Israel | Natural Grassland | Mixed | 2 | Quadcopter | RGB | n/a | 80/80 | 15 to 20 | n/a | 5 collections between April 2018 and April 2020 | Not specified | 600 | 0.25 | n/a |
| (Askari et al., 2019) [98] | Ireland | Experimental Site | Mixed | 1 | Rotary | MS | 30 and 120 | 75/75 | n/a | 2.86 and 11.29 | 6 collections in 2017, 2 collections in 2018 | Mechanical | 126 | n/a | n/a |
| (Barnetson, Phinn and Scarth, 2020) [25] | Australia | Natural Grassland | Mixed | 19 | Hexacopter | RGB and HS | 50 | 85/85 | n/a | 1 | 5 collections 2019 and 1 collection in 2020 | Mechanical | n/a | 0.25 | Electronic RPM |
| (Batistoti et al., 2019) [70] | Brazil | Experimental Site | Mono | 1 | Quadcopter | RGB | 50 | 80/60 | 5 | 1.55 | 7 collections in 2017 and 8 collections in 2018 | Not specified | 66 | n/a | Ruler |
| (Blackburn et al., 2021) [108] | USA | Natural Grassland | Mixed | 19 | Fixed-wing | MS | 122 | 80/75 | n/a | n/a | 1 collection in 2017 | Manual | 190 | 0.01 | n/a |
| (Borra-Serrano et al., 2019) [54] | Belgium | Experimental Site | Mono | 1 | Dodeca-copter | RGB | 30 | 80/80 | 35 | 0.6 | 22 collections in one year | n/a | 154 | 1.05 | RPM |

**Table A2.** *Cont.*

| Reference | Local | Type of Field | Type of Grassland | Number of Sites | UAV Platform | Sensors | Flight Altitude (m) | Overlap, Side Overlap (%) | GCP | GSD (cm/Pixel) | Frequency of Data Collection | Biomass Ground Truth Data | Total Number of Biomass Samples | Biomass Sample Size (m²) | Canopy Height Measurement |
|---|---|---|---|---|---|---|---|---|---|---|---|---|---|---|---|
| (Capolupo et al., 2015) [57] | Germany | Experimental Site | Mono | 1 | Octocopter | HS | 70 | n/a | n/a | 2 | 2 collections in one year | Mechanical | 120 | 12 | RPM |
| (Castro et al., 2020) [96] | Brazil | Experimental Site | Mono | 1 | Quadcopter | RGB | 18 | 81/61 | n/a | 0.5 | 1 collection in 2019 | Mechanical | 330 | 4.5 | n/a |
| (Cunliffe, Brazier and Anderson, 2016) [59] | USA | Natural Grassland | Mixed | 7 | Hexacopter | RGB | 15–20 | 70/65 | 10 to 18 | 0.4 to 0.7 | 1 collection in 2014 | Not specified | n/a | 1 | n/a |
| (da Costa et al., 2021) [101] | Brazil | Natural Grassland | Mixed | 1 | Hexacopter | LiDAR | 100 | n/a | n/a | n/a | 1 collection in 2019 | Manual | 20 | 1 | n/a |
| (De Rosa et al., 2021) [109] | Australia | Grassland Farm | n/a | 2 | Quadcopter | MS | 80 | n/a | n/a | 5 | n/a | RPM calibration | 504 | n/a | n/a |
| (DiMaggio et al., 2020) [48] | USA | Natural Grassland | Mixed | 1 | Quadcopter | RGB | 30, 40, and 50 | 80/80 | 6 | 2.5 | 1 collection in 2018 | Manual | 20 | 0.25 | n/a |
| (Fan et al., 2018) [85] | Japan | Experimental Site | Mono | 1 | Quadcopter | MS | 100 | 50/50 | 13 | 2 | 1 collection in 2016 | Not specified | 36 | 0.25 | Not specified |
| (Franceschini et al., 2022) [52] | Germany | Experimental Site | Mono | 1 | Octocopter | RGB and HS | 30 | n/ | 4 to 8 | RGB = 0.8 and 1.5; Hyper = 7.8 and 15.6 | 2 collections in 2014 and 3 in 2017 | Not specified | 245 | n/a | n/a |
| (Gebremedhin et al., 2020) [66] | Australia | Experimental Site | Mono | 1 | Quadcopter | MS | 20 | 75/75 | 9 | 2 | 3 collections in 2018 | Manual and mechanical | 480 individual plants for calibration and 500 plots for validation | n/a | Ground-based platform (PhenoRover) |
| (Geipel and Korsaeth, 2017) [104] | Norway | Experimental Site | Mono and Mixed | 1 | Octocopter | HS | 50 | n/a | n/a | n/a | 3 collections in 2016 | Manual and mechanical | 120 | n/a | n/a |
| (Geipel et al., 2021) [55] | Norway | Experimental Site | Mixed | 2 | Octocopter | HS | 50 | 80/60 | n/a | n/a | 3 collections in 2016 and 3 collections in 2017 | Mechanical | 707 | ~ 9 | n/a |
| (Grüner, Astor and Wachendorf, 2019) [71] | Germany | Experimental Site | Mixed | 1 | Quadcopter | RGB | 20 | 80/80 | 7 | 0.07 to 0.08 | 4 collections in 2017 | Manual | 192 | 0.25 | Ruler |
| (Grüner, Wachendorf and Astor, 2020) [82] | Germany | Experimental Site | Mixed | 1 | Quadcopter | MS and RGB | 20 and 50 | 100/100 | 8 | 2 and 4 | 3 collections in 2018 | Manual | 144 | 0.25 | n/a |

**Table A2.** *Cont.*

| Reference | Local | Type of Field | Type of Grassland | Number of Sites | UAV Platform | Sensors | Flight Altitude (m) | Overlap, Side Overlap (%) | GCP | GSD (cm/Pixel) | Frequency of Data Collection | Biomass Ground Truth Data | Total Number of Biomass Samples | Biomass Sample Size (m²) | Canopy Height Measurement |
|---|---|---|---|---|---|---|---|---|---|---|---|---|---|---|---|
| (Grüner, Astor and Wachendorf, 2021) [46] | Germany | Experimental Site | Mixed | 1 | Quadcopter | MS and RGB | n/a | n/a | 7 | n/a | 3 collections in 2018 and § collections in 2019 | Not specified | 140 | 0.25 | n/a |
| (Hart et al., 2020) [107] | Switzerland | Grassland Farm | Mixed | 6 | Quadcopter | MS | 50 | 80/80 | 8 | 5 | 4 collections in 2018 | Mechanical | 162 | 6.5 and 1 | n/a |
| (Insua, Utsumi and Basso, 2019) [68] | USA | Grassland Farm | Mixed | 2 | Quadcopter | MS and LiDAR | 100 | 75/75 | n/a | 6 | 2 collections in 2015 and 2 collections in 2016 | Mechanical | n/a | 0.25 | Rapid Pasture Meter (machine) and ruler |
| (Jenal et al., 2020) [102] | Germany | Experimental Site | n/a | 1 | Octocopter | RGB | 30 | n/a | 16 | 4 | 1 collection in one year | Mechanical | 156 | 0.54 × 5.46 m² | n/a |
| (Karila et al., 2022) [115] | Finland | Experimental Site | Mixed | 1 | Quadcopter | RGB and HS | 30 and 50 | n/a | n/a | RGB = 0.8, Hyper = 4 cm | 4 collections in 2017 | Mechanical | 220 | 3.9 (n = 96), ~19.5 (n = 16), 4.5 (n = 108) | n/a |
| (Karunaratne et al., 2020) [65] | Australia | Grassland Farm | Mono | 1 | Quadcopter | MS | 25, 50, 75, and 100 | 80/80 | 10 | 1.74, 3.47, 5.21, 6,94 | 4 collections in 2019 | Mechanical | 101 | 0.25 | n/a |
| (Lee et al., 2015) [84] | Korea | Grassland Farm | Mixed | 1 | Fixed-wing | MS and RGB | 50 | n/a | n/a | 30 | 2 collections in 2014 | Not specified | 56 | 0.03 | n/a |
| (Li et al., 2020) [69] | USA | Experimental Site | Mixed | 1 | Hexacopter | MS and LiDAR | 20 | 85/75 | 7 | 3 | 1 collection in 2019 | Manual | 1320 | Individual Plant | Ruler |
| (Li et al., 2021) [103] | Estonia | Experimental Site | Mixed | 2 | Fixed-wing | MS | 120 | 80/75 | n/a | 10 | 2 collections in 2019 | Not specified | 144 | n/a | n/a |
| (Lussem et al., 2019) [18] | Germany | Experimental Site | Mixed | 1 | Quadcopter | RGB | 25 | 85/85 | 12 | n/a | 9 collections in 2017 | Mechanical | n/a | 4.5 | RPM |
| (Lussem, Schellberg and Bareth, 2020) [16] | Germany | Experimental Site | Mixed | 1 | Quadcopter | RGB | 20 | 90 | 15 | 2 | 2 collections in 2014, 2 collections in 2015, and collections in 2016 | Mechanical | 140 | 15 | RPM |
| (Lussem et al., 2022) [99] | Germany | Experimental Site | Mixed | 1 | Octocopter | RGB and MS | 95 | RGB = 80/80; MS = 75/70 | 15 | RGB = 0.7, MS = 2.3 | 3 collections in 2018 and § collections in 2019 | Mechanical | 832 | 3 | n/a |
| (Michez et al., 2019) [67] | Belgium | Experimental Site | Mixed | 1 | Octocopter | RGB and HS | 50 | 80/80 | 8 | RGB = 2 and MS = 5 | 1 collection in 2017 | Not specified | 40 | 0.09 | LiDAR laser scans |
| (Michez et al., 2020) [45] | Belgium | Experimental Site | Mono | 1 | Quadcopter | MS and RGB | 30 | n/a | 12 | RGB = 1 and MS = 2.5 | 1 collection in 2019 | Mechanical | 29 | 10.5 | Ruler |

**Table A2.** *Cont.*

| Reference | Local | Type of Field | Type of Grassland | Number of Sites | UAV Platform | Sensors | Flight Altitude (m) | Overlap, Side Overlap (%) | GCP | GSD (cm/Pixel) | Frequency of Data Collection | Biomass Ground Truth Data | Total Number of Biomass Samples | Biomass Sample Size (m²) | Canopy Height Measurement |
|---|---|---|---|---|---|---|---|---|---|---|---|---|---|---|---|
| (Näsi et al., 2018) [83] | Finland | Experimental Site | Mixed | 2 | Hexacopter | RGB and HS | 50 and 140 | 73 and 93/65 and 82 | 32 | RGB = 1 and 5 HS = 5 and 14 | 1 collection in 2016 | Mechanical | 32 | 15 | Ruler |
| (Oliveira et al., 2020) [97] | Finland | Experimental Site | Mixed | 4 | Quadcopter | RGB and HS | 30 and 50 | 84–87/65–81 | n/a | HS = 6 and 3, RGB = 0.64 and 0.39 | 3 collections in 2017 | Mechanical | 108 | Different sizes | n/a |
| (Alves Oliveira et al., 2022) [120] | Finland | Experimental Site | Mixed | 1 | Quadcopter | RGB | 50 | n/a | n/a | 1 | 4 collections in 2017 | Mechanical | 96 | ~ 4 | n/a |
| (Pereira et al., 2022) [117] | Brazil | Grassland Farm | Mixed | 1 | Quadcopter | MS | 115 | 75/75 | n/a | 8 | 3 collections in 2019 | Manual | 116 | 1 | n/a |
| (Plaza et al., 2021) [110] | Spain | Grassland Farm | Mixed | 1 | Quadcopter | MS | 43 | n/a | 4 | 3 | 7 collections in 2020 | Not specified | 112 | 0.125 | n/a |
| (Pranga et al., 2021) [60] | Belgium | Experimental Site | Mono | 1 | Hexacopter | MS and RGB | RGB = 40, MS = 30 | 80/70 | 9 | RGB = 0.4, MS = 1.8 | 3 collections in 2020 | Mechanical | 1403 | 7.83 | n/a |
| (Qin et al., 2021) [56] | China | Natural Grassland | Mixed | 82 | Quadcopter | RGB | 20 | n/a | n/a | 1 | 1 collection in 2017 and 1 collection in 2018 | Manual | 300 | 0.25 | n/a |
| (Rueda-Ayala et al., 2019) [37] | Norway | Experimental Site | Mixed | 2 | Quadcopter | RGB | 30 | 90/60 | n/a | n/a | 1 collection in 2017 | Not specified | 20 | 1 | RPM and Ruler |
| (Schucknecht et al., 2022) [94] | Germany | Grassland Farm | Mixed | 3 | Quadcopter and Fixed-wing | MS | Quadcopter = 70, Fixed-wing = 80 | Quadcopter = 80/80, Fixed-wing = 75/75 | 10 | 8.7–12.9 cm; | 1 collection in 2018 | Not specified | n/a | 0.25 | RPM |
| (Schulze-Brüninghoff, Wachendorf and Astor, 2021) [111] | Germany | Natural Grassland | Mixed | 4 | Quadcopter | HS | 20 | 80/60 | 6 | ~20 for spectral images and ~1 for panchromatic band | 3 collections in 2018 | Not specified | 223 | 1 | n/a |
| (Shi et al., 2021) [112] | China | Natural Grassland | Mixed | 1 | Quadcopter | RGB | 40 | n/a | n/a | 1 | 1 collection in 2018 and 1 collection in 2019 | Manual | 432 | 1 | n/a |
| (Shi et al., 2022) [118] | China | Natural Grassland | Mixed | 1 | Quadcopter | RGB | 40 | n/a | n/a | n/a | 1 collection in 2018, 1 collection in 2019, and 1 collection in 2020 | Manual | 648 | 1 | n/a |

**Table A2.** *Cont.*

| Reference | Local | Type of Field | Type of Grassland | Number of Sites | UAV Platform | Sensors | Flight Altitude (m) | Overlap, Side Overlap (%) | GCP | GSD (cm/Pixel) | Frequency of Data Collection | Biomass Ground Truth Data | Total Number of Biomass Samples | Biomass Sample Size (m²) | Canopy Height Measurement |
|---|---|---|---|---|---|---|---|---|---|---|---|---|---|---|---|
| (Shorten and Trolove, 2022) [119] | New Zealand | Experimental Site | Mono | 1 | Quadcopter | RGB | 20 | n/a | n/a | n/a | 2 collections in one 2018 | Not specified | 370 | 1.5 (n = 300), 2.4 (n = 70) | n/a |
| (Sinde-González et al., 2021) [106] | Ecuador | Grassland Farm | Mono | 1 | Quadcopter | RGB | 70 | 80/70 | 8 | 3 | 1 collection in 2018 | Manual | 54 | 0.25 | n/a |
| (Tang et al., 2021) [113] | China | Natural Grassland | Mixed | 4 | Quadcopter | RGB | 10 | 80/65 | 3 | 2.5 | 1 collection in one year | Manual | 623 | n/a | Not specified |
| (Théau et al., 2021) [95] | Canada | Experimental Site | Mixed | 1 | Quadcopter | MS and RGB | 65 | 75/75 | 60 | RGB = 1.7, MS = 6.4 | 2 collections in 2017 | Mechanical | 99 | 0.25 | n/a |
| (Van Der Merwe, Baldwin and Boyer, 2020) [42] | USA | Natural Grassland | Mixed | 11 | Quadcopter | RGB | 40 | 90/85 | n/a | 1 | 1 collection in 2017 and one collection in 2018 | Manual | n/a | 1 | n/a |
| (Viljanen et al., 2018) [26] | Finland | Experimental Site | Mixed | 1 | Quadcopter | RGB and HS | 30 and 50 | RGB = 84/65, MS = 87/81 | 5 | RGB = 0.39 and 0.64; MS = 3 and 5 | 4 collections in 2017 | Mechanical | 96 | ~ 4 | RPM and ruler |
| (Villoslada et al., 2020) [47] | Estonia | Natural Grassland | Mixed | 3 | Fixed-wing | MS | 120 | n/a | 11 | 10 | 1 collection in 2018 | Manual | 140 | 0.09 | n/a |
| (Villoslada et al., 2021) [62] | Estonia | Natural Grassland | Mixed | 9 | Fixed-wing | MS and RGB | 120 | n/a | n/a | RGB = 3.5, MS = 10 | 1 collection in 2019 | Manual | 520 | 0.09 | n/a |
| (Vogel et al., 2019) [73] | Germany | Grassland Farm | Mixed | 1 | Hexacopter | RGB | 100 | 70/70 | n/a | n/a | 1 collection in 2016 | Not specified | 20 | 1 | n/a |
| (Wang et al., 2014) [41] | USA | Natural Grassland | Mixed | n/a | Hexacopter | MS | 5, 20, and 50 | n/a | n/a | 5 m = 0.09; 20 m = 0.36; 50 m = 0.89 | 1 collection in 2013 | Manual | 13 | 0.1 | n/a |
| (Wang et al., 2017) [72] | China | Experimental Site | Mixed | 1 | Octocopter | LiDAR | 10–120 at intervals of 10 m and 120 | n/a | n/a | n/a | 1 collection in 2015 | Manual | 90 | 1 | Ruler |
| (Wengert et al., 2022) [100] | Germany | Grassland Farm And Natural Grassland | Mixed | 4 | Octocopter | HS | 20 | n/a | 6 | 20 | 3 collections in 2018 | Manual | 320 | 1 | n/a |
| (Wijesingha et al., 2019) [44] | Germany | Grassland Farm | Mixed | 3 | Quadcopter | RGB | 25 | 80/80 | n/a | n/a | 8 collections in 2017 | Not specified | 194 | 1 | n/a |
| (Zhang et al., 2021) [51] | China | Experimental Site | Mixed | 1 | Quadcopter | LiDAR | 40–110 (at intervals of 10 m) | n/a | n/a | n/a | 1 collection in 2018 | Manual | 96 | 0.25 | Ruler |

**Table A2.** *Cont.*

| Reference | Local | Type of Field | Type of Grassland | Number of Sites | UAV Platform | Sensors | Flight Altitude (m) | Overlap, Side Overlap (%) | GCP | GSD (cm/Pixel) | Frequency of Data Collection | Biomass Ground Truth Data | Total Number of Biomass Samples | Biomass Sample Size (m²) | Canopy Height Measurement |
|---|---|---|---|---|---|---|---|---|---|---|---|---|---|---|---|
| (Zhang et al., 2022) [81] | China | Natural Grassland | Mixed | 3 | Quadcopter | RGB | 2 | n/a | n/a | n/a | 1 collection in 2018 | Manual | 208 | 0.25 | n/a |
| (Zhang et al., 2018) [8] | China | Natural Grassland | Mixed | 3 | Quadcopter | RGB | 2 and 20 | 70/70 | n/a | 1 | 1 collection in 2017 | Not specified | 75 | 0.25 | n/a |
| (Zhao et al., 2021) [114] | China | Experimental Site | Mono | 1 | Hexacopter | HS | 30 | n/a | n/a | 3 | 1 collection in 2018 | Manual | n/a | 0.09 | n/a |
| (Zhao et al., 2022) [61] | China | Natural Grassland | Mixed | 24 | Fixed-wing | LiDAR | 100~120 | 80/80 | n/a | 1 | 1 collection in one year | Manual | 96 | 1 | Ruler |

**Table A3.** Data analysis methods and essential results of the papers considered in this review.

| Reference | Data Analysis Parameters | | | | Data Analysis Methods and r² from Dry Mass (DM) [1] |
|---|---|---|---|---|---|
| | Spectral Data | Structural Data | Other Data | Terrain Model Source | |
| (Alvarez-Hess et al., 2021) [28] | 5 reflectance bands and 15 spectral indices | n/a | AM plot data only (AP), AM plot plus extreme data (APEX), small polygon data only (SP), and small polygon plus extreme data (SPEX) | n/a | SVR = 0.45 |
| (Adar et al., 2022) [49] | 12 reflectance bands | n/a | Mixed pixels from UAV and satellite, vegetation cover | n/a | SVR = 0.76 |
| (Askari et al., 2019) [98] | 21 spectral indices | n/a | n/a | n/a | PLSR = 0.77, MLR = 0.76 |
| (Barnetson, Phinn and Scarth, 2020) [25] | n/a | Canopy height | n/a | DTMs derived from ground point classification | LR and Automated Machine Learning |
| (Batistoti et al., 2019) [70] | n/a | Canopy height | n/a | DTM derived from ground point classification | LR = 0.74 |
| (Blackburn et al., 2021) [108] | 4 spectral bands and 26 spectral indices | n/a | n/a | n/a | Ridge Estimated Linear models |
| (Borra-Serrano et al., 2019) [54] | 10 spectral indices | 7 canopy height metrics | GDD, ΔGDD between cuts | DTMs from interpolation of ground points and from leaf-off flights | LR = 0.67, MLR = 0.81, PLSR = 0.58, RF = 0.70 |
| (Capolupo et al., 2015) [57] | 4 spectral indices | n/a | n/a | n/a | PLSR = 0.83 |
| (Castro et al., 2020) [96] | n/a | n/a | n/a | n/a | CNNs = 0.88 |
| (Cunliffe, Brazier and Anderson, 2016) [59] | n/a | Canopy height and Canopy volume | Surface cover | DTM derived from ground point classification | LR = 0.95 |
| (da Costa et al., 2021) [101] | n/a | 16 canopy height metrics | Vegetation cover percentage | LiDAR point cloud classification | LR = 0.78 |
| (de Rosa et al., 2021) [109] | NDVI | n/a | n/a | n/a | GAM = 0.60, RF = 0.68 |
| (DiMaggio et al., 2020) [48] | n/a | Mean canopy height and vegetation volume | n/a | DTM by selecting the bare soil lowest point | LR = 0.65 |

**Table A3.** *Cont.*

| Reference | Data Analysis Parameters | | | | Data Analysis Methods and r² from Dry Mass (DM) [1] |
|---|---|---|---|---|---|
| | Spectral Data | Structural Data | Other Data | Terrain Model Source | |
| (Fan et al., 2018) [85] | DN of each band | n/a | n/a | n/a | MLR = 0.84 |
| (Franceschini et al., 2022) [52] | DN of each band | n/a | Variable importance in the projection (VIP) | n/a | PLSR = 0.92 |
| (Gebremedhin et al., 2020) [66] | NDVI | Mean plot height | n/a | n/a | LR = 0.81 |
| (Geipel and Korsaeth, 2017) [104] | NDVI, REIP, and GrassI | Mean plot height | n/a | GPS measurements taken on the ground | PPLSR, MLS and SLR = 0.77 |
| (Geipel et al., 2021) [55] | NDVI and REIP | Mean plot height | n/a | DTM from interpolation of ground points | PPLSR = 0.91; SLR = 0.67 |
| (Grüner, Astor and Wachendorf, 2019) [71] | n/a | Mean plot height | n/a | DTM from interpolation of ground points | LR = 0.72 |
| (Grüner, Wachendorf and Astor, 2020) [82] | 4 spectral bands and 13 spectral indices | n/a | 8 GLCM texture features | n/a | PLSR = 0.76, RF = 0.87 |
| (Grüner, Astor and Wachendorf, 2021) [46] | 13 spectral indices | 15 crop surface height | 8 texture features of each spectral band (4 bands) and 8 texture features of mean CSH, FM, and DM | DTM from TLS data | RF = 0.90 |
| (Hart et al., 2020) [107] | MSI reflectance maps | n/a | Near-infrared reflectance spectroscopy | n/a | LR = 0.29 |
| (Insua, Utsumi and Basso, 2019) [68] | NDVI | Plant height and average ruler sward height | Growth rate | n/a | LR = 0.80 |
| (Jenal et al., 2020) [102] | 12 spectral indices and spectral ground truth | n/a | n/a | n/a | LR = 0.94 |
| (Karila et al., 2022) [115] | RGB and HIS features (spectral bands, several handcrafted vegetation, and spectral indexes) | Canopy height 3D features | n/a | DTM from point cloud classification | Deep pre-trained neural network architectures and CNNs = 0.90 |
| (Karunaratne et al., 2020) [65] | 5 spectral bands and 15 spectral indices | 10 plant height metrics | 4 flight altitudes | DTM from point cloud classification | RF = 0.91 |
| (Lee et al., 2015) [84] | NDVI | n/a | n/a | n/a | LR = 0.77 |
| (Li et al., 2020) [69] | 4 spectral indices | Plant canopy perimeter and canopy height | n/a | DTM from LiDAR data | LR = 0.93 |
| (Li et al., 2021) [103] | 6 spectral indices | n/a | n/a | n/a | RF = 0.9, SVR = 0.89, ANN = 0.99 |
| (Lussem et al., 2019) [18] | 6 spectral indices | Mean sward height and 90th percentile of the sward height | n/a | DTM from leaf-off flight | Bivariate and MLR = 0.73 |
| (Lussem, Schellberg and Bareth, 2020) [16] | n/a | 5 sward height metrics | n/a | DTM from leaf-off flight | LR = 0.86 |
| (Lussem et al., 2022) [99] | 5 spectral bands and 19 spectral indices | 8 sward height metrics | n/a | DTM from leaf-off flight | LR, PLSR, RF and SVM = 0.9 |
| (Michez et al., 2019) [67] | 4 spectral bands and 4 spectral indices | Sward height model | n/a | DTM from LiDAR data | Multivariate models = 0.49 |

**Table A3.** *Cont.*

| Reference | Data Analysis Parameters | | | | Data Analysis Methods and r² from Dry Mass (DM) [1] |
|---|---|---|---|---|---|
| | Spectral Data | Structural Data | Other Data | Terrain Model Source | |
| (Michez et al., 2020) [45] | 14 spectral indices | Sward height model | n/a | DTM from LiDAR data | MLR = 0.74 |
| (Näsi et al., 2018) [83] | 39 spectral bands and 13 spectral indices | 8 canopy height metrics | 2 flight altitudes | DTM from point cloud classification | RF and LR = 0.78 |
| (Oliveira et al., 2020) [97] | 38 spectral bands and 23 spectral indices | 8 canopy height metrics | 2 flight altitudes | DTM from point cloud classification | RF and MLR = 0.97 |
| (Alves Oliveira et al., 2022) [120] | n/a | n/a | n/a | n/a | CNNs = 0.79 |
| (Pereira et al., 2022) [117] | 5 spectral bands and 25 spectral indices | n/a | PlanetScope and Sentinel-2A | n/a | RF = 0.7 |
| (Plaza et al., 2021) [110] | 6 spectral indices | n/a | n/a | n/a | PLSR = 0.782 |
| (Pranga et al., 2021) [60] | 21 spectral indices | 3 canopy height metrics | n/a | DTMs from ground-based GPS interpolation | PLSR, RF and SVM |
| (Qin et al., 2021) [56] | Excess Green Index | Fractional vegetation cover | Pika tunnel length and diameter, pika pile diameter | n/a | LR = 0.446 |
| (Rueda-Ayala et al., 2019) [37] | n/a | Mean plot volume | n/a | n/a | LR = 0.54 |
| (Schucknecht et al., 2021) [94] | 9 spectral bands and 26 spectral indices | In situ bulk canopy height | n/a | n/a | GBM = 0.59, RF = 0.67 |
| (Schulze-Brüninghoff, Wachendorf and Astor, 2021) [111] | n/a | Canopy surface height | Terrestrial laser scanning data | n/a | RF = 0.81 |
| (Shi et al., 2021) [112] | RGBVI | n/a | Bare ground | n/a | LR = 0.88 |
| (Shi et al., 2022) [118] | RGBVI | n/a | Bare ground and mowing ration | n/a | LR |
| (Shorten and Trolove, 2022) [119] | Mean spectral bands for vegetative and soil material | Percent vegetation cover and forage volume | n/a | DTM from interpolation of ground points | LR = 0.66 |
| (Sinde-González et al., 2021) [106] | n/a | Density factor and volume | n/a | DTM from bare ground | Descriptive statistic = 0.78 |
| (Tang et al., 2021) [113] | Band mean and band standard deviation of DN values | n/a | n/a | n/a | PLSR = 0.48 |
| (Théau et al., 2021) [95] | 9 spectral indices | Mean plot volume | Vegetation cover classification | DTMs from ground-based GPS interpolation | LR = 0.94 |
| (Van Der Merwe, Baldwin and Boyer, 2020) [42] | n/a | Canopy height model | n/a | DTM from interpolation of dense point clouds | LR = 0.91 |
| (Viljanen et al., 2018) [26] | 8 vegetation indices | 8 canopy height metrics | n/a | DTM from bare ground and DTM from automatic point classification | MLR = 0.98, RF = 0.97 |
| (Villoslada et al., 2020) [47] | 13 vegetation indices | n/a | n/a | n/a | RF = 0.67 |
| (Villoslada Peciña et al., 2021) [62] | 13 vegetation indices | n/a | n/a | DTM from interpolating the points classified as ground by the Cloth Simulation Filtering algorithm | RF = 0.981 |

**Table A3.** *Cont.*

| Reference | Data Analysis Parameters | | | | Data Analysis Methods and r² from Dry Mass (DM) [1] |
|---|---|---|---|---|---|
| | Spectral Data | Structural Data | Other Data | Terrain Model Source | |
| (Vogel et al., 2019) [73] | Reflectance of red, green, and blue; hue: saturation, value, NDVI, and VARI | n/a | n/a | n/a | LR = 0.8119 |
| (Wang et al., 2014) [41] | NDVI | n/a | n/a | n/a | OLSR = 0.4 |
| (Wang et al., 2017) [72] | n/a | Mean and maximum canopy height and fractional canopy cover | Different flight heights | DTM from LiDAR data | LR = 0.34 |
| (Wengert et al., 2022) [100] | 118 spectral bands | n/a | n/a | n/a | PLSR = 0.45; RF = 0.73, SVR = 0.74, CBR = 0.75 |
| Wijesingha et al., 2019) [44] | n/a | 10 canopy height metrics | n/a | DTM from TLS data | LR = 0.62 |
| (Zhang et al., 2021) [51] | n/a | 3 canopy height metrics and Fractional vegetation cover | n/a | DTM from LiDAR data | MLR = 0.54 |
| (Zhang et al., 2022) [81] | 6 color space indices and 3 vegetation indices | Canopy height model from point clouds | n/a | n/a | RF = 0.78 |
| (Zhang et al., 2018) [8] | n/a | 5 canopy height metrics | n/a | DTM from point cloud ground point classification | LR = 0.76–0.78 |
| (Zhao et al., 2021) [114] | NDVI | n/a | n/a | n/a | PLSR = 0.85 |
| (Zhao et al., 2022) [61] | n/a | 5 canopy height metrics, canopy cover and canopy volume | n/a | n/a | SMR = 0.25 |

[1] SVR = support vector regression; PLSR = partial least squares regression; MLR = multiple linear regression; LR = linear regression; RF = random forest; CNNs = convolutional neural networks; GAM = generalized additive model; PPLSR = powered partial least squares; ANN = artificial neural network; SVM = support vector machines; GBM = gradient boosting machines; OLSR = ordinary least squares regression; CBR = cubist regression.

**Table A4.** Biomass indices used in the papers assessed in this review.

| Vegetation Index | Equation | Papers |
|---|---|---|
| Anthocyanin Reflectance Index 1 [121] | $ARI1 = \left(\frac{1}{G}\right) - \left(\frac{1}{Redge}\right)$ | [28,65] |
| Blue Normalized Difference Vegetation Index [122] | $BNDVI = \frac{(NIR-B)}{(NIR+B)}$ | [99] |
| Canopy Chlorophyll Concentration Index [123] | $CGCI = \frac{\left(\frac{(NIR-Redge)}{(NIR+Redge)}\right)}{NDVI}$ | [28,65,99] |
| Chlorophyll Vegetation Index [124] | $CVI = \frac{NIR}{Green} \times \frac{Red}{Green}$ | [45,47,62,67,117] |
| Colouration Index [125] | $CI = \frac{(R-B)}{R}$ | [60] |
| Datt1 [126] | $Datt1 = \frac{(NIR-RE)}{(NIR-R)}$ | [95] |
| Datt4 [126] | $Datt4 = \frac{R}{G} * Redge$ | [47,62] |
| Difference Vegetation Index [127] | $DVI = NIR - Red$ | [47,62] |

**Table A4.** *Cont.*

| Vegetation Index | Equation | Papers |
|---|---|---|
| Enhanced Vegetation Index [128] | $EVI = 2.5 \times \frac{NIR - Red}{NIR + 6Red - 7.5B + 1}$ | [60,69,117] |
| Enhanced Vegetation Index 2 [129] | $EVI2 = \frac{2.5 \times (NIR - R)}{(NIR + (2.4 \times R))}$ | [28,65,99] |
| Excess Green [130] | $ExG = 2\,G - R - B$ | [26,54,56,60,81,83,97] |
| Excess Green Combined with Canopy Height Model [83] | $ExG + CHM$ | [18,26,97] |
| Excess Green-Red [131] | $ExGR = ExG - ExR$ | [26,54,60,97] |
| Excess Red (Meyer et al., 1998) [132] | $ExR = 1.4\,R - G$ | [26,60,97] |
| GnyLi Vegetation Index [74] | $GnyLi = \frac{R_{910} \times R_{1100} - R_{980} \times R_{1200}}{R_{910} \times R_{1100} + R_{980} \times R_{1200}}$ | [102] |
| Grassland Index [50] | $GrassI = RGBVI + CHM$ | [18,83,97] |
| Green Atmospherically Resistant Vegetation Index [133] | $GARI = \frac{NIR - (G - (B - Red))}{NIR + (G - (B - Red))}$ | [60] |
| Green Chlorophyll Index [134] | $GCI = \left( \frac{NIR}{G} \right) - 1$ | [28,46,60,65,82,83,97,98,102,117] |
| Green Difference Index [135] | $GDI = NIR - G$ | [65] |
| Green Difference Index [136] | $GDI = NIR - R + G$ | [62] |
| Green Difference Vegetation Index [137] | $GDVI = NIR - G$ | [28,47,62,65] |
| Green Index (H = hue, S = saturation, V = brightness) [138] | $GI = 9 \times \left( \frac{H \times 3.14159}{180} \right) + 3 \times S + V$ | [81] |
| Green Infrared Percentage Vegetation Index [139] | $GIPVI = \frac{NIR}{(NIR + G)}$ | [47] |
| Green Leaf Index [140] | $GLI = \frac{(2 \times G - R - B)}{(2 \times G + R + B)}$ | [60,117] |
| Green Normalized Difference Vegetation Index [133] | $GNDVI = \frac{NIR - G}{NIR + G}$ | [29,45–48,61,63,67,83,97,99–101,120] |
| Green Ratio Vegetation Index [137,141] | $GDVI = \frac{NIR}{G}$ | [28,65,98] |
| Green Red Difference Index [127] | $GRVI = \frac{G - R}{G + R}$ | [26,45,47,67,83,97,110] |
| Green Red Edge Vegetation Index | $GRVI_{edge} = \frac{G - Red}{G + Red}$ | [110] |
| Greenness Red Edge | $Gr_{redge} = \frac{G}{Red + G + B}$ | [110] |
| Leaf Chlorophyll Index [142] | $LCI = \frac{(NIR - Redge)}{(NIR - R)}$ | [98] |
| Log Ratio [95] | $LogR^h = log\frac{(NIR)}{(R)}$ | [95] |
| Medium-Resolution Imaging Spectrometer (MERIS) Terrestrial Chlorophyll Index [143] | $MTCI = \frac{(NIR - Redge)}{(Redge - R)}$ | [28,57,65,83,97,98,102,117] |
| Modified Chlorophyll Absorption in Reflectance Index [141] | $MCARI = [((Redge - R) - 0.2) \times (Redge - G)] \times \left( \frac{Redge}{Red} \right)$ | [46,57,60,82,83,97–99,117] |
| Modified Chlorophyll Absorption in Reflectance Index 2 [144] | $MCARI2 = \frac{[1.5[2.5(R_{nir} - R_{red}) - 1.3(R_{nir} - R_{green})]]}{\sqrt{[(2R_{nir} + 1)^2 - (6R_{nir} - 5\sqrt{R_{red}}) - 5]}}$ | [117] |

**Table A4.** *Cont.*

| Vegetation Index | Equation | Papers |
|---|---|---|
| Combined Index with MCARI [145] | $MCARI\_MTVI2 = \frac{(MCARI)}{(MTVI2)}$ | [117] |
| Modified Green Red Vegetation Index [146] | $MGRVI = \frac{(R_G)^2 - (R_R)^2}{(R_G)^2 + (R_R)^2}$ | [26,45,97] |
| Modified Non-Linear Index [147] | $MNLI = \frac{(NIR^2 - R) \times (1 + L)}{NIR^2 + R + L}$ | [98] |
| Modified Simple Ratio [148] | $MSR = \frac{\frac{NIR}{R} - 1}{\sqrt{\frac{NIR}{R}} + 1}$ | [46,47,82,99,103] |
| Modified Soil-Adjusted Vegetation Index [149] | $MSAVI = \frac{2\ NRI + 1 - \sqrt{(2\ NIR + 1)^2 - 8 \times (NIR - Red)}}{2}$ | [47,60,62,83,95,97,99,102,117] |
| Modified Triangular Vegetation Index [144] | $MTVI = 1.2[1.2(NIR - G) - 2.5(R - G)]$ | [83,97–99] |
| Second Modified Triangular Vegetation Index [144] | $MTVI2 = \frac{[1.5[2.5(R_{nir} - R_{red}) - 2.5(R_{nir} - R_{green})]]}{\sqrt{[(2.R_{nir} + 1)^2 - 6R_{nir} - 5\sqrt{(R_{red})}] - 0.5]}}$ | [117] |
| Nitrogen Reflectance Index [150] | $NRI = \frac{(G - R)}{(G + R)}$ | [98] |
| Near-Infrared to Red Edge Ratio [151] | $NIR.RE = \frac{NIR}{RE}$ | [99] |
| Non-Linear Index [152] | $NLI = \frac{(NIR^2 - R)}{NIR^2 + R}$ | [98] |
| Normalized Difference Red Edge [153] | $NDRE = \frac{(NIR - RE)}{(NIR + RE)}$ | [29,45–48,58,63,67,69,83,97,100,101,104,112,120] |
| Normalized Difference Vegetation Index [154] | $NDVI = \frac{NIR - R}{NIR + R}$ | [18,29,42,45–48,56,61,63,66–69,73,84,85,97,99–101,103–106,112,116] |
| Normalized Green Intensity [130] | $NGI = \frac{G}{R + G + B}$ | [60,110] |
| Normalized Green Red Difference Index [127] | $NGRDI = \frac{(G - R)}{(G + R)}$ | [18,45,47,60,99,103,117] |
| Normalized Pigment Chlorophyll Ratio Index [117] | $NPCI = \frac{(R - B)}{(R + B)}$ | [117] |
| Normalized Ratio Index [155] | $NRI = \frac{R_{910} - R_{1200}}{R_{910} + R_{1200}}$ | [102] |
| Optimization Soil-Adjusted Vegetation Index [156] | $OSAVI = \frac{NIR - R}{NIR + R + 0.16}$ | [26,57,83,95,97,99,102,117] |
| Perpendicular Vegetation Index [157] | $PVI = \sin(a)NIR - \cos(a)R$ | [60] |
| Photochemical Reflectance Index (512.531) [158] | $PRI = \frac{R_{512} - R_{531}}{R_{512} + R_{531}}$ | [60,83,97,102] |
| Plant Pigment Ratio Index Red [159] | $PPRI = \frac{(G - B)}{(G + B)}$ | [99] |
| Plant Senescence Reflectance Index [160] | $PSRI = \frac{R - G}{NIR}$ | [98] |
| Ratio Vegetation Index [125] | $RVI = \frac{NIR}{R}$ | [26,45,69,97,102] |
| Red Difference Index [127] | $RDI = NIR - R$ | [28,65] |
| Red Edge Triangular Difference Vegetation Index (core only) [161] | $RTVIcore = 100(NIR - Redge) - 10(NIR - G)$ | [28,47,62,65] |

**Table A4.** *Cont.*

| Vegetation Index | Equation | Papers |
|---|---|---|
| Red Green Blue Vegetation Index Excess [74] | $RGBVI = \frac{(R_G)^2 - (R_B - R_R)}{(R_G)^2 + (R_B - R_R)}$ | [18,45,83,97,99,112,118] |
| Red Edge Chlorophyll Index [134] | $ReCI = \left(\frac{NIR}{Redge}\right) - 1$ | [28,57,65,83,97,98,117] |
| Red Edge Inflection Point [162] | $REIP = 700 + 40 \times \frac{\frac{R_{670} + R_{780}}{2} - R_{700}}{R_{740} + R_{700}}$ | [55,83,97,102] |
| Red Edge Simple Ratio 2 [163] | $SR2 = \frac{NIR}{Redge}$ | [28,46,60,62,65,82,98] |
| Red Edge to Red Ratio [151] | $RE.R = \frac{Redge}{R}$ | [99] |
| Renormalized Difference Vegetation Index [164] | $RDVI = \frac{NIR - Red}{\sqrt{NIR + Red}}$ | [18,46,82,83,97,99,102] |
| Simple Ratio [165] | $SR = \frac{NIR}{R}$ | [65,98,99,117] |
| Soil Adjusted Vegetation Index [156] | $SAVI = \frac{(1+L) \times (NIR - R)}{(NIR + R) + L}$ | [28,46,47,60,65,82,95,98,117] |
| Spectral Ratio 3 [166] | $SR3 = \frac{R}{G}$ | [98] |
| Spectral Ratio 4 [167] | $SR4 = \frac{G}{R}$ | [98] |
| Spectral Ratio 6 [168] | $SR6 = \frac{R}{NIR}$ | [98] |
| Spectral Ratio 7 [169] | $SR7 = \frac{Redge}{NIR}$ | [98] |
| Transformed Vegetation Index 1 [170] | $TVI1 = \frac{NDVI + 0.5}{ABS\,(NDVI + 0.5)} \times \sqrt{ABS\,(NDVI + 0.5)}$ | [95] |
| Triangular Vegetation Index [171] | $TVI = 0.5[120(NIR - G) - 200(R - G)]$ | [117] |
| Triangular Greenness Index [117] | $TGI =$ $-0.5\left[(\lambda_{red} - \lambda_{blue})(R_{red} - R_{green}) - (\lambda_{red} - \lambda_{green})(R_{red} - R_{blue})\right]$ | [117] |
| Transformed Chlorophyll Absorption Reflectance Index [144] | $TCARI = 3[((Redge - R) - 0.2) \times (Redge - G)] \times \left(\frac{Redge}{Red}\right)$ | [117] |
| TCARI Combined Index With OSAVI [144] | $TCARI\_OSAVI = \frac{TCARI}{OSAVI}$ | [117] |
| Visible Atmospherically Resistant Index [172] | $VARI = \frac{G - R}{G + R - B}$ | [18,45,60,73,99,117] |
| Visible Atmospherically Resistant Index Red Edge [172] | $VARIrededge = \frac{(Redge - 1.7R + 0.7B)}{(Redge + 2.3R + 1.3B)}$ | [117] |
| Wide Dynamic Range Vegetation Index [173] | $WDRVI = \frac{\propto NIR - R}{\propto NIR + R}$ | [60] |

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
