# Peer review of "A Review of Estimation Methods for Aboveground Biomass in Grasslands Using UAV"

_remotesensing, doi:10.3390/rs15030639_

Round 1

Reviewer 1 Report

Line 28: research seems to be limited also in terms of plant species and management practices

Line 58: It is not usual to estimate above-ground biomass of grasslands by canopy area. Generally, non-destructive methods are related to visual estimations, height and density measurements, use of capacitance meters, spectral analysis and remote sensing. Some references that could be helpful:

L. ´t Mnnetje and R.M. Jone. Field and laboratory methods for grassland and animal production research. CABI Publishing. 2000.447p.

Alison Davies; R.D.Baker; Sheila A. Grant and A.S. Laidlaw. Sward Measurement Handbook. BGS. 1993. 319p.

Lines 61 to 77. Raising plate meters seems to be widely used for temperate grasslands. There are limitations for its use to estimate above-ground biomass on tropical pastures, rangelands, etc.

Line 148. The term “scale of work” seems to be repeated in the text

Line 278 to 283: It would be interesting to have more information about plant species (legumes, tropical grasses, temperate grasses, shrubs, trees, etc) and pasture management (rotational or continuous grazing; pre or post grazing evaluation, fertilization, etc).

Lines 407 to 419. It would also be important to have studies under different pasture management conditions.

Lines 378 to 504. It is not clear if a sampling processess was used to estimate mean herbage biomass of the hole field or if models were trained baed on the image of just the sample square

Line 462. SfM should be defined

Author Response

We would like to thank the reviewer for the thoughtful and helpful comments. We appreciate the constructive input and believe that the comments improved our manuscript. We carefully inspected each of the reviewer’s comments and answered each of them below.

Reviewer #1:

  • Comment 1, line 28:

“Research seems to be limited also in terms of plant species and management practices.”

Response: We thank the Reviewer for the valuable comment. We agree with the reviewer that there is a limited number of studies investigating these important aspects of management practices and species diversity.

Action: Regarding the number of plant species there are two paragraphs in the text (line 289 to 314) discussing the importance of studying more heterogeneous grasslands. Regarding management practices, we tried to partially cover this topic by adding information on fertilizer. We are also aware that there is still space to be specifically focused on this topic and we recommend it for future studies. Therefore we emphasized on the importance of this aspect as a recommendation that future studies should incorporate more diversified management practices. The added text in lines 1053-1056 in the revised manuscript is as follows:

 “We also strongly recommend that future studies should provide more information on the agronomic aspect of the research area. Detailed overview of soil characteristics, spatial heterogeneity of species distribution, climate, grassland classification and management practices used, enables independent analyses and cross-study comparisons.”

  • Comment 2, line 58:

“It is not usual to estimate above-ground biomass of grasslands by canopy area. Generally, non-destructive methods are related to visual estimations, height and density measurements, use of capacitance meters, spectral analysis and remote sensing. Some references that could be helpful: L. ´t Mnnetje and R.M. Jone. Field and laboratory methods for grassland and animal production research. CABI Publishing. 2000.447p. Alison Davies; R.D.Baker; Sheila A. Grant and A.S. Laidlaw. Sward Measurement Handbook. BGS. 1993. 319p”

Response: We are very grateful to the reviewer for this important contribution. In fact, after checking the suggested references we agreed that canopy area would not be the best term to use in the text.

Action: We removed the term 'canopy area' and replaced it with 'plant density' (line 58). We also added one of the references indicated by the reviewer.

  • Comment 3, lines 61 to 77:

“Raising plate meters seems to be widely used for temperate grasslands. There are limitations for its use to estimate above-ground biomass on tropical pastures, rangelands, etc.”

Response: We are very grateful to the reviewer for this important addition.

Action: We searched for some references to support this information. We found a paper that reports a limitation in the use of RPM for AGB estimation in tropical grasslands and added a sentence with this information. The newly added text (line 76) is now read as:

RPM is also not suitable for grasses with tender erect stems, including some tropical grasses.”

  • Comment 4, line 148:

“The term “scale of work” seems to be repeated in the text

Action: Checked and removed repeated words.

  • Comment 5, line 278 to 283:

“It would be interesting to have more information about plant species (legumes, tropical grasses, temperate grasses, shrubs, trees, etc) and pasture management (rotational or continuous grazing; pre or post grazing evaluation, fertilization, etc).”

Response: We are grateful for the reviewer’s comment. We were planning to add this information (which we already had extracted) to the paper.  However, there was also a limitation in this sense because many papers did not explicitly provide sufficient information about these data. For example, most of the papers did not provide information about climate, management, or classification of the pasture (tropical, temperate, semi-arid). Therefore and to avoid confusion, we decided not to include this classification in the paper.

Action: We emphasized in the recommendations of the paper the importance of this topic to draw the attention of future researchers to put more detailed information on agronomic aspects (added text in lines 1053-1056) as we are mentioned in the reply for Comment 1.

  • Comment 6, line 407 to 419:

“It would also be important to have studies under different pasture management conditions.”

Response: We once again thank the reviewer to point out the importance to have more studies under different pasture management conditions.

Action: As we previously answered for Comment 1, we include in the final recommendations (lines 1053-1056) that future studies should incorporate more diversified management practices.

  • Comment 7, line 378 to 504:

“It is not clear if a sampling processess was used to estimate mean herbage biomass of the hole field or if models were trained baed on the image of just the sample square.”

Response: We are grateful for the reviewer’s comment that indicates that this information was not clear in the text.

Action: We are not sure if we understood the comments correctly. We explained on lines 383-384 that the samples collected in the field are used to establish and evaluate the biomass model derived from UAV images. To make it clearer we added on line 385 that the biomass samples are also used for training. Regarding the sample size used for this process, we explained on lines 423-431 that they are collecting from quadrants or harvesting the entire plot.

  • Comment 8, line 462:

“SfM should be defined.”

Action: Added the definition.

Reviewer 2 Report

This is an excellent comprehensive review paper.  I congratulate the authors on this work.  It is an interesting and important contribution to the literature.

Author Response

We are thankful to the reviewer for general positive feedback on our paper.

Reviewer 3 Report

General comments:

In this review article, authors systematically review published articles on the estimation of aboveground biomass with UAV in grasslands to identify the current landscapes of the studies on the application of UAV on biomass estimation. The work is scientifically sound and can significantly contribute to our understanding of UAV remote sensing in estimating ecosystem productivity and some of the challenges. The paper is well written, with good arguments. However, there are still some minor errors authors can improve upon before the final publication.

Specific Comments:

Line45-line77: These paragraphs are somewhat wordy; you might consider keeping it short and sweet as well.

Line 106: “average field size is low” à “average field size is small”

Line 155: Maybe you can use nested search method so that you can avoid missing out some of the studies.

Figure1: This flow chart may not be necessary to be appeared in the main text. You can put it in the supplementary information.

Line193: How many people did participate in screening process? I recommend authors to conduct cross screening of the papers by at least two people so that you can effectively avoid potential bias.

Line194: Besides, did you use any analytical methods, if so, please provide the descriptions.

Line299: “coefficient of determination” May be this is unnecessary here.

Line307-309: I am wondering, based on the data you have compiled, if it is possible to find the general guidelines for site selection for this type of study to increase the extendibility of the findings or the method proposed? If yes, further discussing these aspects, I believe, is invaluable for future studies.

Line407: Again, I am really curious if it is possible to provide a minimum sampling frequency that can guarantee a certain simulation accuracy range based on the data you have compiled? If yes, this would be useful for future research set up.

Line419: If possible, please provide a chart that shows the accuracy of simulation in different seasons.

Line435-436: Did you conduct quantitative test? if so, please provide the test results. If not, I think you should lower the tone.

Line478: “The mean LiDAR-derived canopy height showed the best linear association 477

with field canopy height measurement (r2 = 0.34)

Really?

Line 514-515: I will consider delete “Because of their distinct properties” since the reasons were given at the end this sentence.

Line 599-600: I suggest to rewrite this sentence.

Line 619, 621, 627: Please unify the abbreviation of the term structure from motion, SFM or SfM? I think the latter is correct.

Line 689, 759: Please give the notification of abbreviation at the first use only.

Line 701: I think you forget to cite relevant studies here.

Line 706: Please unify the number of decimal places in whole text. One or two decimal places? Maybe using two decimal places is better. 0.7, 0.981, 0.96…

Line 762-763: GIS (Geographic Information Systems) platforms

Line 767: Imaginary? imagery or image data.

Line 768: It is better to notify the developers of softwares when the first mention.

Line 769: Python is not a software.

Lines 784-786: So Pix4Dmapper doesn't have this function?

Line 790: Why soil?

Line 802: GCPs (plural)

Line 871: LR?

Lines 873, 874, 875: One notification is enough in one paper.

Line 891: You can use the abbreviation "LR" since it was notified before. Also the same for others.

Line 894: Why use the capital letters?

Line 916: Please notify what are SVM and SVR.

Line 919: What is the difference of RF and RFR?

Line 919: What is CBR?

Line 925: What is PPLSR?

Line 961: What are H98TH and COV?

Line 1000: leaf area index (LAI)

Line 1013: You can also use abbreviations since they were notified.

Line 1024: PLS or PLSR?

Author Response

  • Comment 1.

“In this review article, authors systematically review published articles on the estimation of aboveground biomass with UAV in grasslands to identify the current landscapes of the studies on the application of UAV on biomass estimation. The work is scientifically sound and can significantly contribute to our understanding of UAV remote sensing in estimating ecosystem productivity and some of the challenges. The paper is well written, with good arguments. However, there are still some minor errors authors can improve upon before the final publication.

Response: We are very thankful for the extensive review, and we value the large amount of work and the constructive feedback which helped us to improve the quality of our work. In the revised manuscript, we tried our best to address all the comments and provide detailed justification when changes were implemented.

  • Comment 2, line 45 to line 47.

“These paragraphs are somewhat wordy; you might consider keeping it short and sweet as well”.

Response: We agree with the reviewer that this paragraph is long. We tried our best to keep it short as much as possible. We even now split the paragraph into two separate paragraphs, each focusing on one important topic which are relevant to the construction of the introduction.

Action: We rewrite two sentences and split the paragraph into two shorter paragraphs. The new paragraphs are now read on lines 56-79.

“Ground-based methods for non-destructive measurement of grassland AGB have been studied for decades [15,16]. These approaches estimate AGB using equations relating biomass to measurable biophysical factors such as plant height and plant density [17]. Handheld devices are the most straightforward instruments for measuring these biophysical factors[18]. The most widely used and well-documented ground-based method for the non-destructive measurement of AGB in grasslands is the rising plate meter (RPM) [19]. These instruments measure compressed sward height by integrating sward height and density over a specific area [21]. The ability of RPM-based compressed sward height to estimate AGB grass using regression models is now well established [22–24]. In view of this, farmers use RPM devices to create electromechanical models, which produce accurate and reliable estimates [25].

Despite the benefits of fast and regular assessments RPM method, also has drawbacks, including operator variability and paddock slope. Through uneven and undulating terrain, the RPM method's ability to measure grass height effectively can be impacted, frequently leading to inaccurate measurements due to the RPM base not effectively touching the true ground surface [26]. The RPM also presents limitations when the sward is high and lacks a flat top structure, or when the grass sward is sparse and grows poorly and unevenly [27]. It is also not suitable for grasses with tender erect stems, including some tropical grasses [28]. Additionally, RPM measurements are also point measurements and therefore the within-paddock spatial variability of grassland biomass production is not taken into account because only an average paddock estimate is observed [29].”

  • Comment 3, line 155.

“Maybe you can use nested search method so that you can avoid missing out some of the studies”.

Response: We are grateful for this important recommendation. The entire study selection process was carried out following the PRISMA protocol as recommended by the journal's guidelines (https://www.mdpi.com/journal/remotesensing/instructions). The keywords were also used in different combinations as it is in the nested search method. We believe that the selection of the studies was made in the most comprehensive way possible by the first three co-authors to avoid missing out any studies.

  • Comment 4, Figure 1.

“This flow chart may not be necessary to be appeared in the main text. You can put it in the supplementary information”.

Response: We thank the reviewer for the suggestion. We have based our review on previous review papers published in Remote Sensing Journal. We noticed that in all of them similar flow chart explaining the paper selection was included in the main body of the text.  From our experience, we found that it did help us to better visualize and understand the selection procedure. Considering this aspect we chose to keep the flow chart in the main text.

  • Comment 5, line 193.

“How many people did participate in screening process? I recommend authors to conduct cross screening of the papers by at least two people so that you can effectively avoid potential bias.”

Response: We thank the Reviewer for this important suggestion. The main author of the manuscript was responsible for the screening process. However, the whole process was supported and checked by all the other co-authors. Particularly, the first three co-authors discussed intensively and on a regular base about the selection process right at the beginning of this study.

  • Comment 6, line 194.

“Besides, did you use any analytical methods, if so, please provide the descriptions.”

Response: We thank the Reviewer for this comment. However, we did not use any analytical methods. As the focus of the paper was on the current status of research on this topic which has not been covered so far in the current status literature. We conducted some basic reviews by calculating the average. However, we agree that implementing a comprehensive analytical method would be very interesting as a follow-up study. This review paper could be used as a very good guideline for such analysis. We, therefore, pointed to the value of conducting such study for future follow-up reviews on this topic.

Action:  We added a recommendation for future studies in the lines 1055-1056. The added text in lines 1053-1056 in the revised manuscript as:

 “We also strongly recommend that future studies provide more information on the agronomic aspect of the research area. Details overview of soil characteristics, spatial heterogeneity of species distribution, climate, grassland classification and management practices used, enables independent analyses and cross-study comparisons.”

  • Comment 7, line 299.

““coefficient of determination” May be this is unnecessary here.”

Action: We removed the text.

  • Comment 8, line 307 to 309.

“I am wondering, based on the data you have compiled, if it is possible to find the general guidelines for site selection for this type of study to increase the extendibility of the findings or the method proposed? If yes, further discussing these aspects, I believe, is invaluable for future studies.”

Response: We thank the Reviewer for this significant suggestion. We agree that a general guide for the selection of study sites would be an important contribution to future studies.

But, as we show in figure 2a, there is still quite a big room to expand studies on this topic globally to cover more sites. As we also showed in Figure 2b, the research on this topic is growing exponentially in the last few years. However, this is a complex task that is beyond the scope of our work. We also believe that we still need to fill this mentioned gap before we recommend a general guideline for site selection. This is the reason why we pointed out the importance to conduct studies in a range of grassland fields and throughout different growing seasons (lines 410-422). We hope, however, that our work can serve as a resource for future studies and efforts in this direction.

  • Comment 9, line 407.

“Again, I am really curious if it is possible to provide a minimum sampling frequency that can guarantee a certain simulation accuracy range based on the data you have compiled? If yes, this would be useful for future research set up.”

Response: This is a significant suggestion that we agree would be important for future studies. However, we already pointed out the constraints we faced in terms of the number of studies and experiments and their characteristics (sites, size of the area, species composition, management, objectives, etc.). As we emphasized in the added lines (1053-1056) some of the studies did not provide this information explicitly. Therefore, this kind of recommendation is a challenging task for which we need more information. But we do believe that this review is a first step to achieving this task and that the data set compiled in this work draws the attention of the audience to the value of putting this information in the papers in detail and how it will be useful for future efforts in this direction.

  • Comment 10, line 419.

“If possible, please provide a chart that shows the accuracy of simulation in different seasons.

Response: We thank the Reviewer for this suggestion. We initially thought about producing a chart discussing this topic. Unfortunately, this was not possible firstly because most of the studies worked with data collected on one date only as we pointed out in lines 413-415. Among the studies that were conducted on more than one date, the periods differed (some in different years and not seasons). Therefore, one of our recommendations is that future studies should focus on different seasons to allow a better comparison on how this factor would affect the accuracy of the simulation.

 Comment 11, line 435-436.

“Did you conduct quantitative test? if so, please provide the test results. If not, I think you should lower the tone.”

Response: We thank the reviewer for this important question. As mentioned in another suggestion we did not perform a quantitative test. We, therefore, changed the sentence and adjust it based on the scope of our paper.

Action: We followed the Reviewer's suggestion and rewrote the sentence. The revised text now reads as (on lines 449-453):

“We found comparable results, with an average r2 for manual cutting of 0.68, which was lower than the r2 observed for mechanical harvesting (0.82). These results can also indicate that the number of observations can have more impact on the accuracy of AGB estimation than the collection procedure.”

  • Comment 12, line 478.

“The mean LiDAR-derived canopy height showed the best linear association 477 with field canopy height measurement (r2 = 0.34) Really?”

Response: We agree that this sentence raised confusion. We reconsidered this sentence and realized that this is not directly relevant to the scope of this study. To avoid this confusion, we decided to remove this sentence. We believe that this makes the whole concept clearer.

Action: We checked and considered that the information was not relevant to the discussion and removed the sentence.

  • Comment 13, line 514 to 515.

“I will consider delete “Because of their distinct properties” since the reasons were given at the end this sentence.”

Action: Checked and deleted.

  • Comment 14, line 599 to 600.

“I suggest to rewrite this sentence.”

Response: We took the suggestion and rewrote the sentences to make them more solid and specific.

Action: Although we don't know exactly what should be changed, we have rewritten the sentence in a way that we thought would be more understandable to the readers. The revised text now reads as (on lines 604-606):

“The results gained by DiMaggio et al. [49]  indicate that flying at 50 m height can increase the area that is covered without considerably losing AGB estimation accuracy .

  • Comment 15, line 619, 621and 627

“Please unify the abbreviation of the term structure from motion, SFM or SfM? I think the latter is correct.”

Action: Checked and changed for the right abbreviation (SfM).

  • Comment 16, line 689 and 759.

“Please give the notification of abbreviation at the first use only.”

Action: Checked and removed the repeated notification.

  • Comment 17, line 702.

“I think you forget to cite relevant studies here.”

Action: Checked and added the references.

  • Comment 18, line 706.

“Please unify the number of decimal places in whole text. One or two decimal places? Maybe using two decimal places is better. 0.7, 0.981, 0.96….”

Action: Checked and unified the decimal data by using two decimals.

  • Comment 19, lines 702 - 763.

“GIS (Geographic Information Systems) platforms”

Action: Checked and organized the text as suggested.

  • Comment 20, line 767.

“Imaginary? imagery or image data”

Action: Checked and changed for the correct word ‘imagery’.

  • Comment 21, line 768

“It is better to notify the developers of softwares when the first mention.”

Action: Checked and notified the developers of the software in the first mention in the text.

  • Comment 22, line 769.

“Python is not a software.”

Action: Checked and removed Python from the text.

  • Comment 23, line 784-786.

“So Pix4Dmapper doesn't have this function?”

Response: We thank the reviewer for this important comment. In fact, we consider that this paragraph only informs about some features of the Agisoft software that may also be available in Pix4Dmapper.

Action: We decide to remove the paragraph since it only provides information on tools available in a software package without making any real contribution to the discussion.

  • Comment 24, line

“Why soil?”

Response: We thank the reviewer for this important inquiry. Although we stated in the text that the GCPs are placed on the soil, the information is not exactly correct, since the GCPS can also be placed in other visible sites.

Action: We checked and changed soil for ‘a visible site’.

  • Comment 25, line

“GCPs (plural)”

Action: Checked and changed GCP for GCPs (plural).

  • Comment 26, line

“LR?”

Action: Checked and changed for LR.

  • Comment 27, line 873, 874, 875.

“One notification is enough in one paper”

Action: Checked and removed repeated notifications.

  • Comment 28, line

“You can use the abbreviation "LR" since it was notified before. Also the same for others.”

Action: Checked and removed repeated notifications.

  • Comment 29, line

“Why use the capital letters?”

Action: Checked and removed capital letters.

  • Comment 30, line

“Please notify what are SVM and SVR.”

Action: Checked and added the notification.

  • Comment 31, line

“What is the difference of RF and RFR?”

Action: Checked and corrected the term to RF.

  • Comment 32, line

“What is CBR?”

Action: Checked and added the notification.

  • Comment 33, line

“What is PPLSR?”

Action: Checked and added the notification.

  • Comment 34, line

“What are H98TH and COV?”

Action: Checked and added the notification.

  • Comment 35, line

“leaf area index (LAI)”

Action: Checked and added the notification.

  • Comment 36, line

“You can also use abbreviations since they were notified.”

Action: Checked and changed for the abbreviation form.

  • Comment 37, line

“PLS or PLSR?.”

Action: Checked and added the notification.